# Prediction of stimulus-independent and task-unrelated thought from functional brain networks

Aaron Kucyi [1✉], Michael Esterman[2,3], James Capella[4], Allison Green[5], Mai Uchida[5,6], Joseph Biederman[5,6], John D. E. Gabrieli[4,7], Eve M. Valera[6,8,9] & Susan Whitfield-Gabrieli[1,9]

Neural substrates of "mind wandering" have been widely reported, yet experiments have varied in their contexts and their definitions of this psychological phenomenon, limiting generalizability. We aimed to develop and test the generalizability, specificity, and clinical relevance of a functional brain network-based marker for a well-defined feature of mind wandering—stimulus-independent, task-unrelated thought (SITUT). Combining functional MRI (fMRI) with online experience sampling in healthy adults, we defined a connectome-wide model of inter-regional coupling—dominated by default-frontoparietal control subnetwork interactions—that predicted trial-by-trial SITUT fluctuations within novel individuals. Model predictions generalized in an independent sample of adults with attention-deficit/hyperactivity disorder (ADHD). In three additional resting-state fMRI studies (total $n = 1115$), including healthy individuals and individuals with ADHD, we demonstrated further prediction of SITUT (at modest effect sizes) defined using multiple trait-level and in-scanner measures. Our findings suggest that SITUT is represented within a common pattern of brain network interactions across time scales and contexts.

[1] Department of Psychology, Northeastern University, Boston, MA, USA. [2] National Center for PTSD & Neuroimaging Research for Veterans Center (NeRVe), Veterans Administration Boston Healthcare System, Boston, MA, USA. [3] Department of Psychiatry, Boston University School of Medicine, Boston, MA, USA. [4] Department of Brain and Cognitive Sciences, Massachusetts Institute of Technology, Cambridge, MA, USA. [5] Clinical and Research Program in Pediatric Psychopharmacology and Adult ADHD, Massachusetts General Hospital, Boston, MA, USA. [6] Department of Psychiatry, Harvard Medical School, Boston, MA, USA. [7] Athinoula A. Martinos Imaging Center at the McGovern Institute for Brain Research, Massachusetts Institute of Technology and Harvard University, Cambridge, MA, USA. [8] Department of Psychiatry, Massachusetts General Hospital, Charlestown, MA, USA. [9] These authors contributed equally: Eve M. Valera, Susan Whitfield-Gabrieli. ✉email: a.kucyi@northeastern.edu

Humans spend a large proportion of their waking lives engaged in mind wandering[1,2], often defined as self-generated experiences that are decoupled from immediate environmental inputs and current tasks at hand[3–5]. The tendency to mind wander varies substantially within and between individuals, and this variability can have severe, broad mental health consequences in conditions such as attention-deficit/hyperactivity disorder (ADHD) and Alzheimer's disease[6–8]. Thus, understanding the nature of mind wandering and its neural basis has emerged as a central goal within cognitive and clinical neuroscience[4,6,9,10].

Within the last decade, neuroimaging combined with online experience sampling—wherein people are intermittently prompted to report their current thoughts—has become a well-established, powerful method to link brain dynamics with mind-wandering episodes[11–16]. Initial functional MRI (fMRI) studies using this and related techniques confirmed an association between mind wandering and default mode network (DMN) activation[11,17,18], a finding now also supported by fMRI[19,20] and intracranial electrophysiology[21] evidence of increased DMN activity preceding behavioral lapses (despite a nuanced role in task engagement[13,14,20]). Subsequent research has additionally revealed the critical role of distributed, dynamic, network-level interactions within and beyond the DMN[6,9,12,13,22–25], including functional coupling patterns of the frontoparietal control network (FPCN)[25,26] and primary sensory/motor regions[13,24,27–30]. Further lesion and neurostimulation evidence suggest potential causal roles of core DMN[31–35] and FPCN[36,37] regions. Despite this multimodal evidence for key roles of distinct regions and their interplay, mind wandering has been defined in distinct manners and in different contexts across experiments[3], limiting generalizability of findings. It remains unknown whether a common functional network pattern could provide a specific, generalizable marker of any specific aspect of mind wandering within and between individuals, across distinct clinical groups, and across multiple contexts.

Predictive modeling of neuroimaging data[38,39] has emerged as a promising tool for developing generalizable functional network markers of various cognitive and affective functions[40–43]. Data-driven, multivariate approaches within a predictive modeling framework could be fruitful in the study of mind wandering, given the hypothesized role of interactions within and between multiple distributed brain networks[6,9,10]. Indeed, prior research suggests that multivariate features, based on within- and between-DMN functional connectivity, carry predictive information about task-unrelated thought in healthy adults (at the within-dataset, single-task context level)[12,44]. Establishing broader generalizability of a network-based model would have significant implications for understanding the brain bases for conditions such as ADHD, where excessive mind wandering is strongly associated with clinical outcome[8]. In addition, a predictive model may have broad value for the interpretation of functional neuroimaging data because mind wandering may be an important, but typically unexplained, source of brain variability within any given experimental or clinical session[9,45].

Here we aimed to identify and test the generalizability, specificity, and clinical relevance of a brain network-based marker for a well-defined feature that is considered central to mind wandering—stimulus-independent, task-unrelated thought (SITUT)[3–5]. First, using experience sampling in healthy adults, we defined a functional network-based model that predicted trial-by-trial fluctuations in SITUT within novel individuals. We tested the model's validity and its specificity to the construct of SITUT relative to related, but not equivalent constructs (e.g., sustained attention) and characterized the model's features in relation to functional neuroanatomy previously linked to mind wandering. Next, we tested whether the same model-predicted trial-wise SITUT within adults with ADHD who reported an increased frequency of SITUT during experience sampling. Finally, in three independent resting-state fMRI (rs-fMRI) datasets from distinct cohorts, we tested whether the model predicted individual trait and state mind wandering, as defined with multiple distinct measures of daily life and in-scanner SITUT.

## Results

**A functional connectivity pattern predicts SITUT within healthy adults.** To identify a functional network-based marker of SITUT, we analyzed data from healthy adults who participated in an fMRI study with online experience sampling at Massachusetts General Hospital (MGH). Participants completed the Gradual-Onset Continuous Performance Task[20] (gradCPT) while receiving intermittent thought probes every 44–60 s. The paradigm uniquely allowed us to capture unique neural predictors of SITUT while accounting for behavioral performance (reaction time) fluctuations[14]. Upon each thought probe (36 total trials), participants provided subjective ratings of task-related focus on a graded scale, with 0 indicating fully "on-task" and 100 indicating fully "off-task." We retained for initial analysis only those participants who reported that their "off-task" reports were due to task-unrelated thoughts that were also stimulus-independent (17 participants and a total of 612 trials; see "Methods"). Specifically, the included participants retrospectively reported that their "off-task" ratings were more strongly due to SITUT ($M \pm SD$ rating on 7-point Likert scale: $5.5 \pm 1.1$) than to external distractions (EDs: $M \pm SD = 2.1 \pm 1.2$) or to thoughts about task-related interferences (TRIs: $M \pm SD = 2.8 \pm 1.7$) (SITUT vs. EDs: $P = 2.1 \times 10^{-6}$; SITUT vs. TRIs: $P = 1.1 \times 10^{-4}$; Wilcoxon rank-sum tests).

We applied connectome-based predictive modeling (CPM)[46] to model and predict trial-wise, intra-individual fluctuations in SITUT from functional connectivity within a leave-one-participant-out (leave 36 trials out) cross-validation framework. For each thought probe (trial), we computed the functional connectivity matrix within the 30-s window (28 fMRI frames) preceding probe onset based on a 268-node whole-brain functional atlas (Shen268) (Fig. 1a). Within each cross-validation fold (comprising 16 participants), we identified all node pairs (edges) exhibiting a suprathreshold-level positive or negative correlation with within-participant normalized SITUT ratings. Based on the positive minus negative edge sum scores for each trial, we constructed a linear model to predict SITUT based on all 576 trials (i.e., 16 participants × 36 trials) within a given cross-validation fold. For the held-out one participant (36 trials), we applied this linear model to compute predicted trial-wise SITUT, which we correlated with observed SITUT rating and then compared the correlation value with a null distribution (Fig. 1b).

Predicted versus observed SITUT correlations were significantly greater than mean permutation test-based null values in held-out participants ($M \pm SD$ within participant $r = 0.11 \pm 0.16$, $P = 0.019$, Wilcoxon signed-rank test; MSE = $1.19 \pm 0.15$) (Fig. 1c). This provided evidence that our functional network model was predictive of SITUT within novel individuals at the within-dataset level (i.e., establishing internal validation) and with an overall effect size that was on par with that typically found for functional connectivity-based prediction of self-report outcomes[47]. The edges contributing to the model (hereafter referred to as "SITUT-CPM" masks) included 258 and 139 edges, respectively, positively and negatively associated with SITUT. These edges were distributed widely throughout the brain, with high-degree nodes (i.e., nodes involved in multiple contributing edges) situated in prefrontal, parietal and temporal cortices and cerebellum (Fig. 1d).

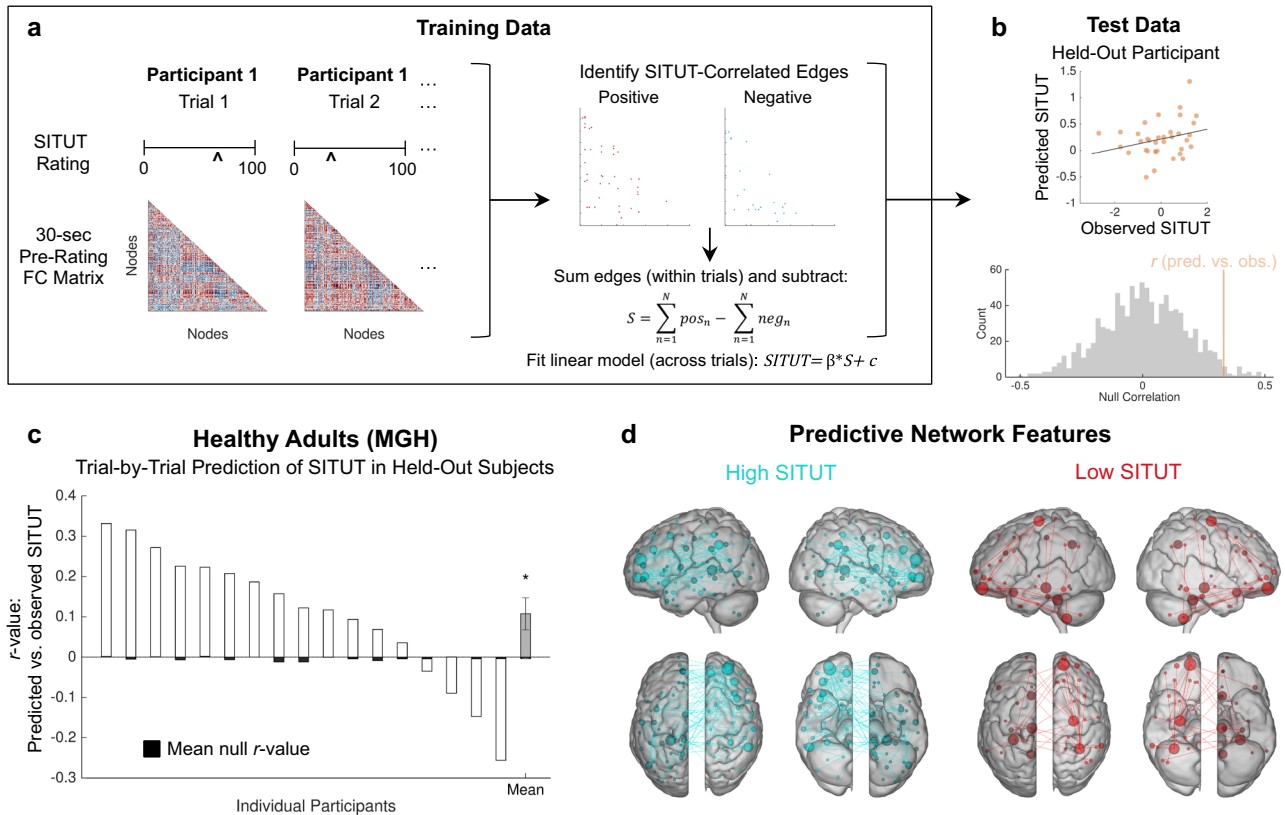

**Fig. 1 Functional connectivity-based predictive modeling of trial-wise SITUT fluctuations within healthy adults. a** A schematic of the analysis pipeline. Within training data, SITUT ratings and pre-rating functional connectivity matrices (based on a 268-node whole-brain atlas) were extracted for each trial. Red and blue, respectively, indicate positive and negative functional connectivity. Edges that were correlated with SITUT rating were identified at a threshold of $P < 0.01$ (uncorrected), and a summary score was obtained for each trial based on a subtraction of positive and negative edge sums. **b** A linear model, based on summary scores, was used to correlate 36 predicted versus observed SITUT ratings in a held-out participant (top). Subsequently, that correlation value (indicated with orange line) was compared with a null distribution of $r$ values derived from 1000 permutations of shuffled SITUT ratings (bottom). **c** Predicted versus observed SITUT correlations, and mean null correlations, within each held-out participant. The gray bar indicates the mean across individuals (i.e., mean across white bars; $n = 17$). At the group level, predicted versus observed correlations were significantly greater than mean null correlations ($P = 0.019$, two-sided, Wilcoxon signed-rank test). Source data are provided as a Source data file. **d** Edges strongly contributing positively (blue) and negatively (red) to the predictive model. A degree threshold of 6 was applied; i.e., nodes involved in at least 6 contributing edges are displayed. Error bars indicate standard error of the mean. SITUT stimulus-independent, task-unrelated thought. *$P < 0.05$.

**Validity and specificity of the SITUT-CPM.** We next performed confirmatory analyses to assess the validity and specificity of the SITUT-CPM. Despite performance of fMRI preprocessing steps that aim to limit the impact of nuisance factors on functional connectivity (as done here; see "Methods"), frame-wise head motion can still influence observed relationships between functional connectivity and behavior[48,49]. However, we found that SITUT-CPM compared to null model prediction remained significant when using partial correlations within held-out participants in each cross-validation fold, controlling for mean frame-wise head motion ($r_{partial} = 0.091 \pm 0.15$, $P = 0.039$). In addition, though our main analysis was based on data preprocessed with an aCompCor[50] noise-correction pipeline, prediction remained significant when using an alternative ICA-AROMA[51] approach to control for head motion ($r = 0.10 \pm 0.14$, $P = 0.0086$). When using an alternative 300-node functional atlas (Schaefer300), trial-wise SITUT-CPM network strength was similar to that initially estimated with the Shen268 atlas (within-participant trial-wise correlation between Shen268- versus Schaefer300-derived network strengths: $r = 0.87 \pm 0.054$), and prediction of SITUT remained significant ($r = 0.12 \pm 0.23$, $P = 0.044$).

Given that SITUT and sustained attention are inter-related, but not equivalent constructs[52], we tested for specificity of SITUT-CPM predictions using a simultaneous online behavioral measure, reaction time (RT) variability, indicative of sustained attention[14,20,52] and previously shown to be positively correlated with SITUT[5,14,53–55]. In our participants, self-reported SITUT was positively associated with pre-thought probe (30-s mean) RT variability across trials ($F_{1,612} = 8.70$, $P = 0.0033$; $F$ test on linear mixed-effects model), suggesting behavioral validation of subjective reports but also implying overlapping variance between the two measures. However, SITUT-CPM compared to null model prediction remained significant when controlling for pre-thought probe RT variability in held-out participants (partial $r_{partial} = 0.10 \pm 0.16$, $P = 0.040$).

Moreover, we compared our newly defined SITUT-CPM to the previously defined Sustained Attention CPM (SA-CPM) shown to be predictive of gradCPT behavioral performance within and between individuals[41,56]. The SITUT-CPM and SA-CPM masks did not show significant edge overlap (Fig. S1a). Trial-wise SITUT-CPM versus SA-CPM network strengths were significantly negatively correlated at the group level ($r = -0.13 \pm 0.21$, $P = 0.031$) (Fig. S1b), suggesting an expected relationship of higher expression of a brain marker for SITUT associated with lower expression of a marker for sustained attention. However, when controlling for SA-CPM network strength, SITUT-CPM compared to null model prediction of observed mind wandering remained significant ($r_{partial} = 0.096 \pm 0.16$, $P = 0.028$), suggesting

independence between the behavioral relevance of the SITUT-CPM versus SA-CPM. Moreover, SA-CPM predictions were not significantly associated with observed SITUT ($r = -0.073 \pm 0.16$, $P = 0.093$), despite an expected negative trend.

In addition, although SITUT and creative thinking may draw from similar cognitive processes[57], the SITUT-CPM showed independence from a previously published CPM[40] based on individual differences in creative ability (Cr-CPM). Though there was some significant edge overlap between the SITUT-CPM and Cr-CPM (Fig. S1c), trial-wise network strengths from the two CPMs were independent from one another ($r = -0.029 \pm 0.18$, $P = 0.94$) (Fig. S1d), and SITUT-CPM prediction remained significant when controlling for Cr-CPM strength ($r_{partial} = 0.11 \pm 0.17$, $P = 0.031$). Collectively, these findings provide evidence for validity and specificity of the SITUT-CPM.

**Functional neuroanatomical basis of the SITUT network.** For improved interpretation of the functional neuroanatomical basis of patterns contributing to the SITUT-CPM, we examined relationships with well-described functional networks previously linked to mind wandering. Based on the Schaefer300 atlas, with each node assigned to 1 of 7 standard Yeo-Krienen networks[58], we quantified the number of SITUT-CPM mask edges belonging to each intra- or inter-network pair. Among edges positively correlated with SITUT, DMN within- and between-network connections contributed most strongly (Fig. 2a). The top 5 network pairs contributing to positive edges were DMN-FPCN, DMN-DMN, sensorimotor-visual, DMN-sensorimotor, and sensorimotor-dorsal attention. The DMN-FPCN contribution, which was greatest (61 total pairs), was explained by anti-correlated activity during task-focused trials and lack of anti-correlation during SITUT trials (Fig. 2b). This finding aligns with prior theoretical and empirical research supporting a role of DMN-FPCN interactions in various forms of mind wandering[6,11,59].

Among edges negatively correlated with mind wandering, DMN-sensorimotor network (SMN) pairs (30 total) contributed most strongly (see Fig. 2c for other contributions). The DMN-SMN contribution was explained by positive correlation during task-focused trials and anticorrelation during SITUT trials (Fig. 2d). This finding aligns with the notion that during mind wandering, the DMN engages its capacity to draw from information unrelated to immediate sensory input[24,60].

Given that within-network functional heterogeneity is a key consideration within current theoretical models of mind wandering[6,10], we examined the specific contributions of subnetworks based on the Yeo-Krienen 17-network atlas. This revealed that positively contributing DMN-FCPN edges were largely explained by specific interactions between DMN$_A$ (as labeled within the Schaefer atlas), a subnetwork including the medial prefrontal and posteromedial cortices (consistent with the DMN "core"[61]), and FPCN$_A$[62], a subnetwork anchored in rostrolateral prefrontal cortex (Fig. 2e). Negatively contributing DMN-SMN edges were largely explained by interactions between the DMN$_A$ and SMN$_A$, a subnetwork comprised of primary somatosensory and motor cortices (Fig. 2f). In summary, although SITUT-CPM predictions were based on a complex, distributed pattern of interacting networks, key components associated with increased SITUT were a) decreased DMN$_A$-FPCN$_A$ anticorrelation; and b) decreased DMN$_A$-SMN$_A$ correlation (Fig. 2g).

Further univariate tests showed that among all network pairs (mean across regions) within the Yeo-Krienen 7-network atlas, only DMN-FPCN was significantly associated with SITUT at the group level when stringently correcting for all comparisons

(Fig. S2a). Moreover, trial-wise SITUT-CPM network strength showed the greatest correlation with mean DMN-FPCN connectivity than with all other network pairs (Fig. S2b). To further interrogate the relative importance of DMN-FPCN pairs, we performed a version of CPM with DMN-FPCN edges only rather than using all edges in the whole-brain (Schaefer300) atlas. This CPM analysis revealed significant prediction of SITUT, based on DMN-FPCN edges alone, within held-out participants ($r = 0.14 \pm 0.19$, $P = 0.015$). Moreover, when virtually 'lesioning' DMN-FPCN edges from the whole-brain atlas, prediction significance was reduced ($r = 0.079 \pm 0.21$, $P = 0.15$). These results collectively suggest that DMN-FPCN edges were sufficient and necessary for prediction performance within the sample of healthy adults used to define the SITUT-CPM (i.e., for internal validation). Thus, our subsequent analyses examine whether the SITUT-CPM, based on all nodes, or based on DMN-FPCN edges only (hereafter termed "SITUT-CPM$_{DMN-FPCN}$"), was generalizable to other cohorts and contexts (external validation).

**External validation #1: Intra-individual prediction of mind wandering in ADHD.** Having established the neurophysiological plausibility of the SITUT-CPM, we turned next to establishing external validity within independent datasets. First, we examined generalizability in adults with ADHD ($n = 20$) who participated in the same gradCPT experience sampling study as the healthy adult group (i.e., with the same MRI scanner at MGH). We hypothesized that (a) the SITUT-CPM—generated within healthy adults—would predict intra-individual SITUT ratings in an independent ADHD sample (i.e., demonstrating external validation); and (b) given the association between increased mind-wandering frequency and ADHD symptoms[8,63], the SITUT-CPM network strength would be over-expressed, on average, within ADHD compared to healthy participants throughout task performance.

Correlations between SITUT-CPM prediction (based on the healthy participant model) and observed SITUT in participants with ADHD were significantly greater than mean null values ($r = 0.045 \pm 0.086$, $P = 0.028$) (Fig. 3a), suggesting generalizability from the healthy individuals to the individuals with ADHD. To further confirm this generalizability, we also independently generated a CPM to predict intra-individual SITUT within the ADHD, rather than healthy, sample (using leave-one-participant out), which revealed significant internal validation within ADHD ($r = 0.12 \pm 0.15$, $P = 0.0032$; MSE $= 1.21 \pm 0.22$) as well as significant external validation in healthy adults ($r = 0.072 \pm 12$, $P = 0.015$) (Fig. 3b). Thus, network-based predictors of intra-individual SITUT were overlapping between the two groups. However, the simpler SITUT-CPM$_{DMN-FPCN}$ (generated in healthy adults) did not yield significant prediction in ADHD ($r = 0.0053 \pm 0.091$, $P = 0.74$), potentially suggesting overfitting and an important role of edges beyond DMN-FPCN in facilitating generalizability (as such, we do not further test SITUT-CPM$_{DMN-FPCN}$ generalizability).

As anticipated, SITUT ratings during the gradCPT were significantly higher in the ADHD compared to healthy control (HC) group [$M \pm SD$ on 100-point scale: ADHD ($55.3 \pm 12.5$), HC ($34.8 \pm 18.8$); $P = 9.6 \times 10^{-5}$, Wilcoxon rank-sum test] (Fig. 3c). The ADHD group also showed lower ratings of awareness to where their attention was focused ($M \pm SD$ on 100-point scale: ADHD ($50.3 \pm 15.3$), HC ($67.6 \pm 16.7$); $P = 0.0016$). In addition, at the neural level, mean SITUT-CPM network strength was significantly greater in the ADHD than in HC group [$M \pm SD$: ADHD ($9.54 \pm 8.98$), HC ($4.98 \pm 6.09$); $P = 0.039$] (Fig. 3d). In contrast, SA-CPM network strength was not significantly different between groups [$M \pm SD$: ADHD ($-12.5$

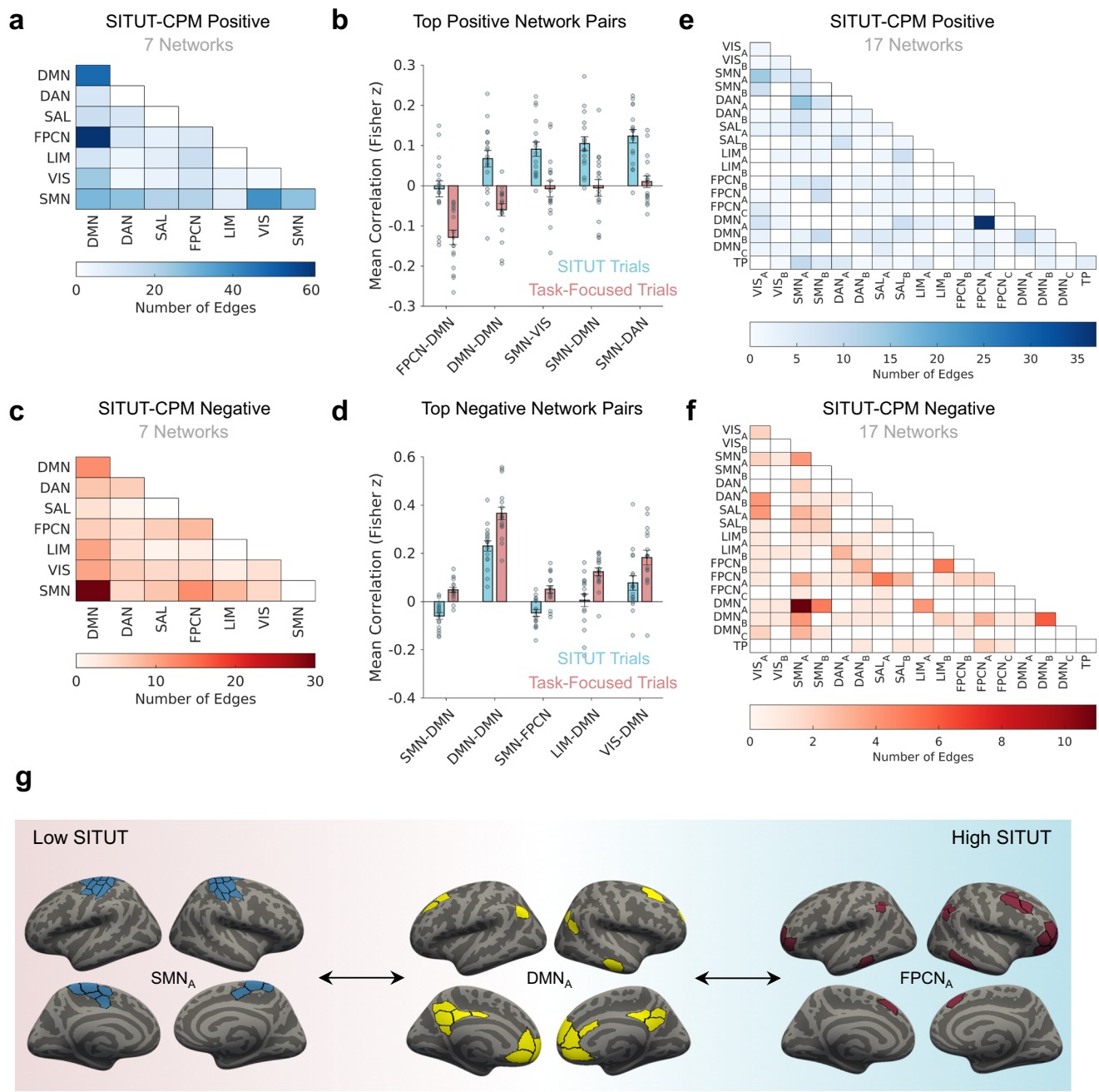

**Fig. 2 Functional neuroanatomical basis of the SITUT-CPM network. a** The number of edges, among those within the SITUT-CPM positive mask, assigned to each within- or between-network pair based on the Schaefer300 and Yeo-Krienen 7-network atlases. **b** The mean correlation between CPM network-pair regions as a function of SITUT rating, shown for the top five positive network-pair features. "SITUT" and "Task-Focused" trial values were derived from a median split of trials, based on SITUT ratings, within each participant ($n = 17$ participants). Error bars indicate standard error of the mean. **c** Same as **a**, except for the SITUT-CPM negative mask. **d** Same as **b**, except for edges negatively correlated with SITUT ($n = 17$ participants). **e** Same as **a**, except for the Yeo-Krienen 17-network atlas. **f** Same as **b**, except for the Yeo-Krienen 17-network atlas. **g** Summary schematic of major inter-network pairs contributing to the CPM-based SITUT network. DAN dorsal attention network, DMN default mode network, FCPN frontoparietal control network; LIM limbic network, SAL salience network, SMN sensorimotor network, TP temporal-parietal network, VIS visual network, SITUT stimulus-independent, task-unrelated thought, CPM connectome-based predictive model. Source data are provided as a Source data file.

$\pm 44.6$), HC ($-25.3 \pm 37.5$); $P = 0.37$]. Taken together, these findings confirm (a) the SITUT-CPM is sensitive to intra-individual SITUT ratings in an independent sample and different group of individuals (ADHD), and (b) increased SITUT in individuals with ADHD during task performance (coupled with lower awareness of SITUT) is mirrored in increased SITUT-CPM network strength at the neural level.

**External validations #2–4: Inter-individual prediction of trait SITUT from rs-fMRI.** We have so far established a functional network model that carries predictive information about intra-individual SITUT, measured online during task performance, within healthy individuals and individuals with ADHD. In the next analyses (external validations 2–4), we focused on whether the SITUT-CPM is sensitive to trait-level measures of SITUT

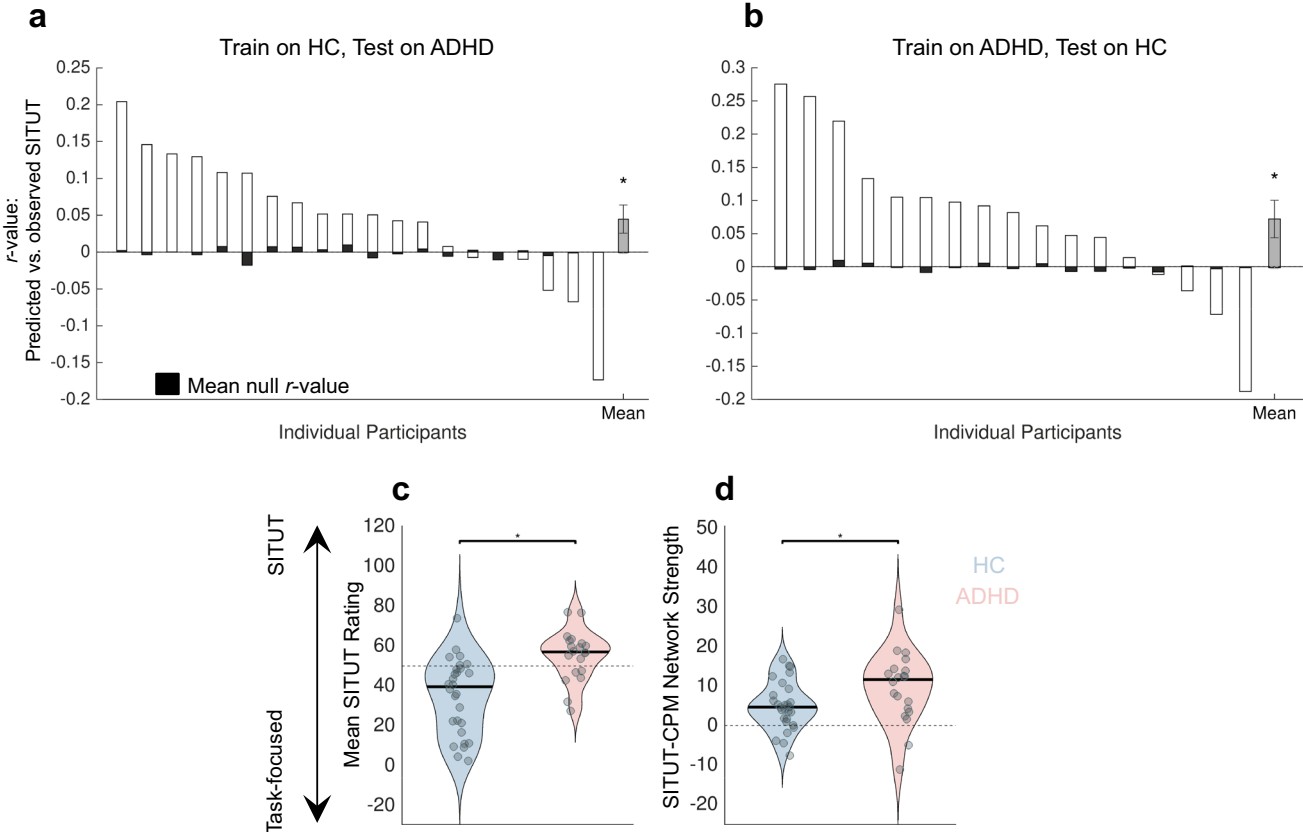

**Fig. 3 The SITUT-CPM is sensitive to intra-individual mind wandering in adults with ADHD. a** Predicted versus observed SITUT correlations, and mean null correlations, within each held-out participant with ADHD, based on the SITUT-CPM generated in healthy control (HC) participants. At the group level ($n = 20$ individuals with ADHD), predicted versus observed correlations were significantly greater than mean null correlations (*$P = 0.028$, two-sided, Wilcoxon signed-rank test). The gray bar indicates the mean across individuals (i.e., mean across white bars). **b** Same as **a** but for held-out HC participants ($n = 17$) based on a CPM generated to predict SITUT within ADHD participants (*$P = 0.015$, two-sided, Wilcoxon signed-rank test). **c** Mean SITUT ratings were significantly greater in the ADHD compared to HC group (*$P = 9.6 \times 10^{-5}$, two-sided, Wilcoxon rank-sum test). **d** SITUT-CPM network strength, averaged across gradCPT runs within each participant, was significantly greater in the ADHD compared to HC group (*$P = 0.039$, two-sided, Wilcoxon rank-sum test). Error bars indicate standard error of the mean. Solid black lines in **c** and **d** indicate median. SITUT stimulus-independent, task-unrelated thought, CPM connectome-based predictive model. *$P < 0.05$. Source data are provided as a Source data file.

**Table 1 Participant characteristics for three rs-fMRI datasets included in analyses of trait SITUT.**

|  | Group | N (included) | Females/males | Age (M ± SD) | Participants excluded (high motion) | Trait SITUT assessment |
|---|---|---|---|---|---|---|
| Superstruct | Healthy | 911 | 531/380 | 21.61 ± 2.86 | 15 | DDFS |
| Leipzig | Healthy | 144 | 64/80 | 32.43 ± 11.97 | 44 | MW-D, MW-S |
| MIT | ADHD | 49 | 25/24 | 30.69 ± 7.49 | 10 | MWQ |

*DDFS* Daydreaming Frequency Scale; *MW-D* Mind Wandering Deliberate Scale; *MW-S* Mind Wandering Spontaneous Scale; *MWQ* Mind Wandering Questionnaire.

based on rs-fMRI, which measures spontaneous brain activity in an awake, "task-free" condition[64]. We tested whether the SITUT-CPM was predictive of individual differences in SITUT tendencies in three rs-fMRI datasets (Table 1). As our SITUT-CPM definition was based on state SITUT measures (i.e., predictions at the intra-individual level), these analyses were motivated by the premise that high trait SITUT individuals could be more likely to engage in SITUT during rs-fMRI scans. In three rs-fMRI datasets, we examined SITUT-CPM predictions versus observed scores on three distinct questionnaires (Table 1) that participants completed offline. Though the questionnaires include distinct items compared to one another, each one captures both stimulus-independence and task-unrelatedness of thoughts[65–67], thus providing the opportunity to test generalizability and robustness

of the SITUT-CPM, regardless of varying aspects that are unrelated to the construct of interest[38].

**External validation #2: Superstruct dataset (healthy adults).** First, we examined predictions of trait mind wandering within 911 healthy adults from the Brain Genomics Superstruct project. These participants completed one 6-min run of rs-fMRI and, separately, the Daydreaming Frequency Scale (DDFS) to characterize individual mind wandering tendency. We found a significant positive correlation between DDFS score and SITUT-CPM prediction from resting-state functional connectivity (Spearman's $\rho = 0.074$, $P = 0.025$) (Fig. 4a), which remained significant when controlling for frame-wise head motion (partial Spearman's $\rho = 0.080$, $P = 0.016$) and age (partial $\rho = 0.072$, $P = 0.029$). Though the effect size was

**Fig. 4 SITUT-CPM predictions from resting-state fMRI correlate with individual differences in daydreaming frequency scale scores (Superstruct dataset). a** Significant positive correlation between SITUT-CPM-predicted versus observed scores on the daydreaming frequency scale within the Superstruct rs-fMRI dataset. The trend line (for visualization purposes) is based on locally weighted regression fitting with a second-order polynomial. Statistical testing was performed based on Spearman's rank correlation (coefficient and two-sided $p$ value shown). Source data are provided as a Source data file. **b** Correlations (ordered by rank) between SITUT-CPM prediction and 67 behavioral and self-report outcomes in the Superstruct dataset. Among all outcomes, daydreaming frequency (denoted as "MindWandering_Freq") showed the highest correlation with model prediction. Light to dark blue scale indicates lower to higher absolute correlation value. See ref. [68] for a phenotype legend of labels shown. SITUT stimulus-independent, task-unrelated thought, CPM connectome-based predictive model.

modest, we examined the specificity of DDFS prediction relative to a comprehensive battery of 67 distinct behavioral and self-report individual outcomes acquired among participants in the dataset. Among this entire set of individual outcomes, SITUT-CPM prediction showed the strongest correlation with DDFS score, including in comparison to correlation with performance on a Flanker task purported to more generally capture attentional and inhibitory control abilites[68] (Fig. 4b). Further demonstrating specificity, SA-CPM-based predicted sustained attention was not significantly correlated with DDFS (Spearman's $\rho = 0.022$, $P = 0.50$).

**External validation #3: Leipzig dataset (healthy adults).** Second, we examined SITUT-CPM predictions of trait mind wandering within 144 healthy adults scanned at the University of Leipzig. These participants completed four ~15-min rs-fMRI runs (i.e., had ~10× more data per participant than the Superstruct dataset) and later completed the Mind Wandering Deliberate (MW-D) and Spontaneous (MW-S) scales. We found a significant positive correlation between MW-D score and SITUT-CPM prediction from resting-state functional connectivity (Spearman's $\rho = 0.17$, $P = 0.043$) (Fig. 5a), which remained significant when controlling for frame-wise head displacement (partial Spearman's $\rho = 0.18$, $P = 0.032$) and age (partial Spearman's $\rho = 0.17$, $P = 0.038$). There was a similar correlation trend for MW-S (Spearman's $\rho = 0.15$, $P = 0.068$; head motion-controlled partial Spearman's $\rho = 0.16$, $P = 0.055$; age-controlled partial Spearman's $\rho = 0.16$, $P = 0.050$) (Fig. 5b). Providing evidence for specificity, predicted sustained attention from the SA-CPM was not significantly associated with MW-D (Spearman's $\rho = -0.0018$, $P = 0.98$) or MW-S (Spearman's $\rho = 0.010$, $P = 0.90$).

**External validation #4: MIT dataset (ADHD adults).** As a further test of generalizability to rs-fMRI data in a clinical context, we investigated sensitivity to trait SITUT in an independent ADHD sample. Previous findings indicate that individuals with ADHD with high Mind Wandering Questionnaire (MWQ) scores

(>23) exhibit increased severity of ADHD symptoms[8]. Thus, we classified 49 adults with ADHD into high ($n = 31$) versus low ($n = 18$) MWQ subgroups within a cohort of patients that underwent rs-fMRI (7 min each) at the Massachusetts Institute of Technology (MIT). These two groups did not show significant differences in head motion ($P = 0.43$, Wilcoxon rank-sum test, $M \pm SD$ of $0.092 \pm 0.027$ mm for high MWQ participants, $0.088 \pm 0.027$ mm for low MWQ participants), age ($P = 0.097$; $M \pm SD$ of $32.0 \pm 7.5$ for high MWQ participants, $28.2 \pm 7.0$ for low MWQ participants), sex (high MWQ: 18 females, 13 males; low MWQ: 7 females, 11 males), or ADHD medication status (high MWQ: 27.6% using medication; low MWQ: 25% using medication). However, across the entire group, MWQ score was positively associated with inattention and hyperactivity symptom severity within the past 6 months (inattention: $\rho = 0.34$, $P = 0.017$; hyperactivity: $\rho = 0.34$, $P = 0.017$) and showed a weaker, but also positive association with diagnostic symptom severity (inattention: $\rho = 0.23$, $P = 0.11$; hyperactivity: $\rho = 0.23$, $P = 0.12$), in line with prior findings from a larger cohort of patients with ADHD[8].

We found that SITUT-CPM network strength during rs-fMRI was significantly greater in participants with ADHD in the high compared to low MWQ subgroup ($P = 0.043$) (Fig. 6). A continuous correlation within the whole group between SITUT-CPM network strength versus MWQ score revealed a concordant result ($\rho = 0.28$, $P = 0.055$), which remained consistent when controlling for frame-wise head motion (partial $\rho = 0.29$, $P = 0.042$) and age (partial $\rho = 0.28$, $P = 0.058$). In contrast, SA-CPM network strength was not correlated with MWQ score ($\rho = -0.01$, $P = 0.94$).

We further examined whether SITUT-CPM and SA-CPM network strengths were related to ADHD symptom severity (rather than MWQ scores). As shown in Supplementary Table 1, SITUT-CPM network strength was not significantly predictive of symptoms (though trends were positive), while SA-CPM showed its strongest positive relationship with diagnostic inattention severity. These results suggest that SITUT-CPM network strength was more strongly associated with a SITUT-based outcome (MWQ score)

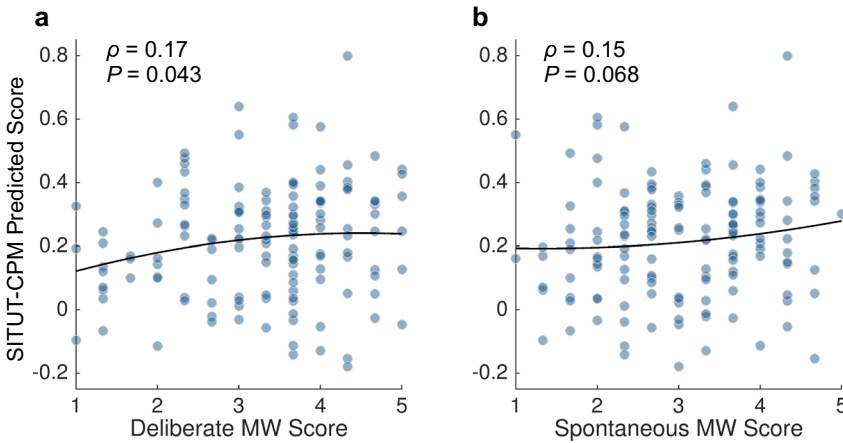

**Fig. 5 SITUT-CPM predictions from resting-state fMRI data correlate with individual differences in deliberate and spontaneous mind wandering questionnaire scores (Leipzig dataset). a** Significant positive correlation between SITUT-CPM-predicted versus observed scores on the Mind Wandering Deliberate scale within a dataset collected at the University of Leipzig. **b** Same as **a** but for the scores on the Mind Wandering Spontaneous scale. Trend lines (for visualization purposes) are based on locally weighted regression fitting with a second-order polynomial. For both **a** and **b**, statistical testing was performed based on Spearman's rank correlation (coefficients and two-sided *p* value shown without correction for multiple comparisons). SITUT stimulus-independent, task-unrelated thought, CPM connectome-based predictive model, MW mind wandering. Source data are provided as a Source data file.

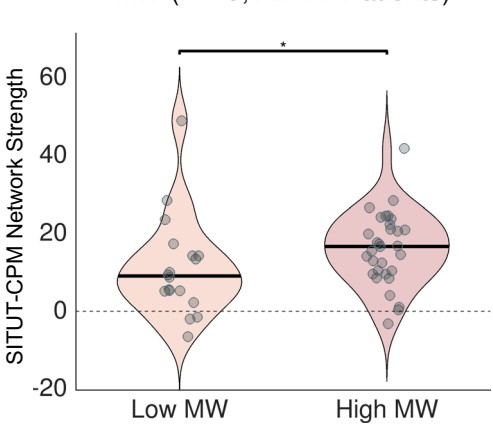

**Fig. 6 SITUT-CPM network strength during rs-fMRI is greater in adults with ADHD with high compared to low Mind Wandering Questionnaire scores.** Individuals were classified as people showing high levels of mind wandering and people showing low levels of mind wandering based on the Mind Wandering Questionnaire. SITUT-CPM network strength was computed based on the dot product of SITUT-CPM suprathreshold edges (see "Methods" and Fig. 1) and connectivity values in a given resting-state scan ($P = 0.043$, two-sided, Wilcoxon rank-sum test). Solid black horizontal line indicates median. SITUT stimulus-independent, task-unrelated thought, CPM connectome-based predictive model, MW mind wandering. *$P < 0.05$. Source data are provided as a Source data file.

than with ADHD symptom severity, whereas SA-CPM network strength was more strongly associated with symptom severity (inattention) than with MWQ score. Taken together, in addition to SITUT-CPM prediction of state dynamics within adults with ADHD (Fig. 3), these findings provide evidence that the SITUT-CPM distinguished individuals with high versus low SITUT tendency within an independent ADHD cohort.

**State-dependent dynamics of the SITUT-CPM during rs-fMRI.** Our findings so far suggest that the SITUT-CPM is sensitive to both state SITUT within individuals and trait SITUT between individuals. One possible explanation for this generalizability is that high SITUT individuals are more likely to spontaneously engage in SITUT during rs-fMRI scans. If so, then the SITUT-CPM should be sensitive to time-varying SITUT states, within individuals, between rs-fMRI scans.

To examine this possibility, we computed SITUT-CPM network strength across four consecutive ~15 min rs-fMRI runs in the Leipzig dataset, where participants completed retrospective experience sampling after each run. We focused on a rating for the item "my thoughts involved my surroundings" to estimate the stimulus-independence of thoughts during a given run. This "surroundings" rating gradually decreased over time across rs-fMRI runs and showed a significant interaction with run number ($F_{1,502.74} = 4.68$, $P = 0.03$; $F$ test on linear mixed-effects model) (Fig. 7, black). Post hoc comparisons revealed a significant decrease in "surroundings" rating following run 2 compared to 1 ($P_{FDR} = 0.016$; Wilcoxon signed-rank test), and the overall trend suggested that the rating remained relatively stable following this post-run 1 decrease. This 'tuning out' over time was expected, given that SITUT intensity generally increases gradually following a task's onset[69]. Importantly, thoughts about surroundings were dissociable from self-reported wakefulness, which did not decrease over time (Fig. S3).

At the neural level, SITUT-CPM network strength also showed a significant interaction with rs-fMRI run number ($F_{1,419.73} = 14.56$, $P = 0.00016$; $F$ test on linear mixed-effects model). The general trend indicated an increase in network strength following the first run (Fig. 7, blue), with post hoc tests revealing significant increases for run 3 compared to 1, 3 compared to 2, and 4 compared to 1 ($P_{FDR} = 0.0027$, $0.0039$, and $0.035$, respectively; Wilcoxon sign rank tests). Thus, across the time scale of ~1 h, thought content gradually diverged from immediate surroundings while SITUT-CPM network strength increased within individuals. However, network strength was not related to run-to-run changes in the "surroundings" rating ($F_{1,669.65} = 0.22$, $P = 0.64$; $F$ test on linear mixed-effects model), suggesting that experience sampling at the time scale of every ~15 mins was not fully sensitive to the SITUT-CPM (or vice versa).

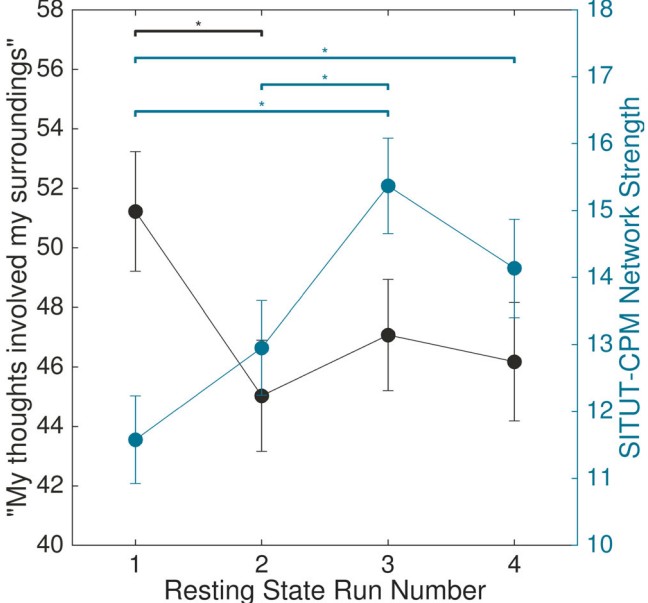

**Fig. 7 Across rs-fMRI scans, thoughts about surroundings decrease while SITUT-CPM network strength increases over time.** In the University of Leipzig dataset, experience sampling following each ~15 min rs-fMRI run (within one session) revealed that self-reported thoughts involving surroundings decreased (i.e., stimulus-independence of thoughts increased) after the first run and then remained relative stable (black; participants included in analyses: $n = 164$ for run 1; $n = 164$ for run 2; $n = 164$ for run 3; $n = 162$ for run 4). Conversely, SITUT-CPM strength increased after the first run (blue; participants included in analyses: $n = 140$ for run 1; $n = 165$ for run 2; $n = 140$ for run 3; $n = 146$ for run 4). Solid dots indicate mean, and error bars indicate standard error of the mean. Statistical tests were corrected for multiple comparisons ($*P_{FDR} < 0.05$, two-sided, Wilcoxon rank-sum test). Significant increases were found for run 3 compared to 1, 3 compared to 2, and 4 compared to 1 ($P_{FDR} = 0.0027$, 0.0039, and 0.035, respectively). SITUT stimulus-independent, task-unrelated thought, CPM connectome-based predictive model. Source data are provided as a Source data file.

## Discussion

Here we developed a neuroimaging-based model (SITUT-CPM) of functional brain networks that reliably predicted trial-by-trial fluctuations in stimulus-independent, task-unrelated thought—a key feature of mind wandering—within novel individuals. The SITUT-CPM was based on a distributed functional connectivity pattern that varied across 30-s windows and included major contributions of within- and between-network interactions strongly linked to mind wandering (operationalized in various ways) in prior theoretical and empirical research[6,9,12,24,59]. The SITUT-CPM predictions were generalizable across healthy and ADHD adults and were sensitive to the increased frequency of SITUT in ADHD. Moreover, in three independent resting-state fMRI samples, including both healthy individuals and individuals with ADHD, we found consistent evidence for further generalizability (with modest effect sizes) to predictions of trait and state mind wandering, as defined with multiple distinct measures of daily life and in-scanner SITUT. Though prediction effect sizes were modest, they were in line with typical functional network-based predictions of self-report outcomes in large samples[47] and were striking given the various differences among datasets (e.g., online task/experience sampling versus rs-fMRI of varying duration as well as different MRI scanners/protocols and outcome measures). Our findings suggest that SITUT is represented within a common pattern of brain network interactions across multiple time scales and contexts. We discuss how our findings fit within and advance the neuroscience of mind wandering, clinical implications of these findings, and broad implications for rs-fMRI biomarker development.

**Neural basis of mind wandering**. Our findings build on prior work demonstrating neural substrates of mind wandering, typically operationalized based on task-unrelated thought[11,12,70,71], but sometimes including measurements of stimulus-independence[22,72], multidimensional features of thought contents[13,29] or other single features[15]. Focusing on a SITUT-based neural model, we were able to demonstrate generalizability and robustness to multiple distinct definitions of mind wandering that are typically used in the field. We found that the distributed nature of interactions within and between distinct brain networks was critical to generalizability of SITUT-CPM predictions. Many of the intra- and inter-network interactions that strongly contributed to SITUT-CPM predictions are consistent with findings from prior neuroimaging, electrophysiological, neurostimulation, and lesion studies of mind wandering. Some less dominant, but contributing, interactions were not necessarily anticipated (Fig. 2), and an understanding of their potential significance requires further study.

Increased coupling (or reduced antagonism) between DMN and FPCN, a dominant SITUT-CPM feature, has previously been associated with goal-directed cognition with an internal focus[73] and was hypothesized to underlie the top-down control needed to deliberately maintain a stream of thought[6,25,26,59]. We show that DMN-FPCN interactions during SITUT are largely characterized by decreased anticorrelation between the DMN and a specific FPCN subsystem encompassing the rostrolateral prefrontal cortex, anterior inferior parietal lobule, and pre-supplementary motor area (FPCN_A). Regions in FPCN_A, but not the closely interdigitated regions in the FPCN_B, show selective coupling with the DMN during a wide variety of contexts[62]. Moreover, intracranial EEG evidence suggests increased theta band coupling between the DMN and FPCN_A, but not FPCN_B, during attention to self-generated thought when faced with exteroceptive input[74]. These converging findings provide support for neurophysiological plausibility of the SITUT-CPM, which was based on key contributions of DMN-FPCN_A interactions. Moreover, this evidence highlights specificity of DMN interactions that underlie SITUT versus sustained attention, a related, but not equivalent, set of cognitive processes typically operationalized based on behavioral performance[52]. Intra- and inter-individual differences in sustained (externally-oriented) attention have been associated with coupling between the DMN and dorsal attention network[19,21,75] rather than FPCN; moreover, we found little anatomical overlap and only minimally shared variance between the SITUT-CPM and the previously published sustained attention CPM[41].

Another key SITUT-CPM component was increased anticorrelation between the DMN and primary sensory and motor regions in SMN_A. Compared to other cortical association networks, the DMN is uniquely situated at the furthest geodesic distance and longest connectivity path from primary sensory regions (including SMN), potentially suggesting a capacity to draw from information untethered to immediate sensory input[76]. Thus, DMN-SMN_A antagonism may reflect the perceptual decoupling (i.e., dampened processing of sensory input) and decreased motor preparedness that is characteristic of SITUT[24,60]. The coupling patterns of primary sensory/motor regions with association networks, including DMN and FPCN, have previously been linked to individual differences in specific contents of experience during mind wandering[27,28].

Within-DMN connectivity contributed both positively and negatively to the SITUT-CPM, consistent with evidence for within-DMN functional heterogeneity[61,77]. Prior research suggests that individual differences in trait mind wandering are both positively and negatively associated with within-DMN functional connectivity[18,23,26,33]. The SITUT-CPM takes this functional heterogeneity into account, which likely contributed to the success of predictions across contexts.

Importantly, our model was based on functional connectivity-based features only. This methodological choice was based on theoretical (i.e., hypothesized role of intra- and inter-network dynamic communication[6,9]) and practical considerations. Consistent with predictive modeling studies in other domains[40,41,56], our network-based approach facilitated model testing across a broad set of external datasets (e.g., rs-fMRI scans in which baseline regional activation cannot be readily inferred). In prior research, functional connectivity, regional activity, and pupil diameter carried complementary predictive information about task-unrelated thought[12]. It is likely that inclusion of such measures and further multimodal recordings could improve model performance[44,78]; however, large datasets that would be needed to test external validity of such richer models are currently unavailable.

**Clinical implications for ADHD.** Our findings from two separate ADHD cohorts highlight SITUT-CPM generalizability to a clinical context and offer novel neurophysiological evidence for previously identified, clinically relevant ADHD subgroups[8]. We speculate that patients with ADHD with high, compared to low, SITUT would more directly benefit from a therapy that targets brain network interactions contributing to the SITUT-CPM. Given that high mind wandering in ADHD is associated with more severe clinical features (e.g., inattention, hyperactivity, executive function, emotional dysregulation, quality of life[8]), a treatment targeting SITUT could hold promise toward improving multiple, rather than single, symptoms/outcomes.

Importantly, although we demonstrated that DMN-FPCN functional connectivity was, overall, a key feature contributing to SITUT-CPM predictions in healthy participants, predictions in ADHD were largely based on non-DMN-FPCN network pairs. A possible reason for this group difference is that the ADHD, relative to healthy, group reported lower awareness of where their attention was focused (in the MGH sample). This decreased awareness may indicate spontaneous, as opposed to deliberate, SITUT, a feature previously linked to ADHD symptoms[63,79]. It is possible that increased DMN-FPCN coupling more closely reflects deliberate, compared to spontaneous mind wandering[6], and as such, DMN-FPCN coupling corresponded to SITUT only in healthy participants in our study because those individuals were experiencing SITUTs more deliberately than were the participants with ADHD. Notably, various rs-fMRI studies have shown a relationship between ADHD and reduced resting-state DMN anticorrelation with other networks, including the FPCN[80–82], as well as with dampened task-evoked FPCN activation and DMN deactivation[83]. Thus, we hypothesize that patients with ADHD have an altered baseline level of DMN-FPCN interaction (associated with spontaneous SITUT) and that other, distributed network interactions account for psychological features that are common between spontaneous and deliberate SITUT.

Beyond ADHD, the SITUT-CPM should be tested in other psychiatric and neurological conditions that exhibit alterations in mind wandering (see ref. [6]), such as rumination in depression and anxiety. Growing evidence suggests that changes in SITUT characterize a wide range of diseases and could be an early marker of Alzheimer's disease[7]. One study suggests that the relationship of mind wandering tendency with DMN-FPCN and within-DMN functional connectivity is altered in patients with neurodegeneration[33]. Thus, further study of SITUT-CPM generalizability, and potential alternative models in disease, could have far-reaching implications within various clinical contexts.

**Implications for resting-state fMRI biomarkers.** Over the past two decades, rs-fMRI has become a standard method to investigate the self-organization of intrinsic activity into large-scale functional networks[64,84]. Even though the unconstrained rs-fMRI paradigm inherently involves SITUT[9,18], individual-specific topographic patterns of correlated brain activity are highly reliable across repeat measurements[85,86]. As such, rs-fMRI shows promise as a tool for identifying neural markers of individual trait-like qualities, such as biomarkers of disease[87]. However, intrinsic network estimates can be influenced by an individual's current mental state. Investigators have extensively characterized how functional networks reconfigure during cognitive task performance relative to wakeful rest[86,88]. In contrast, despite growing interest in rs-fMRI network dynamics across various time scales, less is known about the potential impact of intra-individual mental state fluctuations within and between rs-fMRI scans[89,90].

Here we have begun to uncover how SITUT may impact rs-fMRI network estimates, as we found that the same functional network pattern that was predictive of state SITUT within individuals was predictive of trait SITUT/mind wandering at rest. The latter finding may be driven by (a) a trait-related, enduring change in the brain's intrinsic network architecture; (b) increased SITUT at the time of rs-fMRI scanning within high trait SITUT individuals; or (c) a combination of trait and state SITUT. We suggest that increased SITUT at the time of scanning is at least a likely contributor, given that within individuals, we found increased SITUT-CPM network strength across rs-fMRI runs that coincided with self-reported thought contents drifting away from immediate surroundings. In future studies, our approach could be extended to isolate an rs-fMRI "signal" component that represents relatively stable, intrinsic functions from a more state-dependent "noise" component that represents SITUT[9]. Taken together, our findings call for the need to further investigate how mind wandering affects rs-fMRI intrinsic network biomarkers[45].

**Limitations.** A key limitation concerns the specific task context (gradCPT) in which we defined the SITUT-CPM. During the gradCPT, visual detection of scenes (mountains and cities) engages brain regions (e.g., medial temporal and retrosplenial[20,91]) that could overlap, or dynamically interact, with key networks involved in SITUT (e.g., DMN). Thus, during time windows prior to thought probes, functional connectivity patterns are not only a product of an individual's level of SITUT but also various other processes such as scene detection. As such, inter-regional interactions involved in SITUT could have been missed due to masking with specific functions associated with ongoing gradCPT performance. Though the SITUT-CPM was sensitive to predictions generated from resting-state data, effects were modest, and it remains possible that a different task context for model generation would result in improved out-of-sample prediction performance.

A related concern surrounds our use of 30-s pre-thought probe windows to compute functional connectivity for training and testing of the CPM. This analysis choice was based on our specific task design as well as prior evidence demonstrating that functional connectivity computed from similar durations can reliably distinguish distinct cognitive tasks[92,93] and mind wandering levels[12]. Though brain activity within the seconds

that immediately precede thought probes are most relevant to self-reported experiences[14,94], our selection of a ~30-s window was based on the trade-off between the need to capture activity time-locked to experiences and the need to estimate functional connectivity within a window long enough to provide confidence in the sampling of correlations[95]. However, there is no a priori reason to suspect that 30 s corresponds to the time course of cognitive processes that unfold during any individual episode of SITUT[9]. In future work, applications of methods sensitive to instantaneous edge co-fluctuations[96], as well as unsupervised methods sensitive to frame-wise network 'states'[97], could be valuable toward better resolving the time course of network dynamics that underlie SITUT.

In our experience sampling study, online thought probes assessed task focus and awareness of task focus, while retrospective ratings assessed the overall degree to which reports of off-task focus were due to SITUT, external distractions, or task-related interferences. Though online and retrospective ratings typically show good correspondence[98], retrospective ratings could have been associated with decreased relative accuracy in our study, as participants may not have precisely remembered the proportion of time spent focused off-task for each of the three respective categories. Prior studies, using thought probes to assess these three categories online, suggest that participants may fluctuate from trial to trial in their reasons for reporting off-task focus and that the distinct categories have different neural correlates[72]. Moreover, multidimensional experience sampling has revealed that distinct thought categories recur over time and are linked to different functional connectivity patterns across individuals[13,27,29]. Future work adopting such a multidimensional approach to online thought probes could better clarify cross-trial heterogeneity within individuals and potentially lead to development of more specific neural models of SITUT components that are either individual-specific or generalize across individuals.

**Summary and future directions**. We have demonstrated that SITUT is represented within a common pattern of brain network interactions across multiple time scales and contexts. Our model-based approach can continue to advance the basic and clinical understanding of mind wandering through further testing across novel conditions, comparisons with alternative models, and continuous refinement based on newly emerging evidence.

## Methods

**Overview**. We conducted our analyses with multiple datasets that were acquired as part of four distinct research projects. The Massachusetts General Hospital (MGH) sample included an fMRI experience sampling study of healthy and ADHD cohorts. The Superstruct and Leipzig samples included rs-fMRI data from healthy adults who completed questionnaires assessing trait SITUT. The Massachusetts Institute of Technology (MIT) sample included rs-fMRI data from adults with ADHD who completed a questionnaire that assesses trait SITUT.

We first performed analyses within the MGH healthy adult sample to define a 'normative' functional brain network model of SITUT at the within dataset/cohort level (internal validation) that could be then used to test in other datasets/cohorts (external validation). We next tested the generalizability of the model to (1) within-participant predictions of SITUT in ADHD (MGH sample); (2) between-participant predictions of trait SITUT (Superstruct sample); (3) between-participant predictions of trait SITUT (Leipzig sample); and (4) distinguishing high and low SITUT subgroups in ADHD (MIT sample). Moreover, we assessed whether the model was sensitive to changes in stimulus-independence of experience across rs-fMRI runs (based on post-run experience sampling) over time (Leipzig sample). We selected datasets for analysis based on inclusion of questionnaires designed to assess mind wandering in a manner that captures trait-level SITUT. Demographic information for the three rs-fMRI datasets is shown in Table 1.

## Cohorts

*MGH study procedures*. Participants in the MGH study included healthy adults ($n = 29$) and adults diagnosed with ADHD ($n = 20$) who provided written informed consent for procedures approved by the Partners Human Research Institutional Review Board. Exclusion criteria for healthy adults included current mood, psychotic, anxiety (excluding simple phobias) or attentional disorder, current use of psychotropic medication, full-scale IQ <80, neurological disorder or sensorimotor handicap that could reasonably impact the outcome, current alcohol or substance abuse/dependence, and claustrophobia or other contraindications for MRI. The same criteria were applied to ADHD participants, except that a diagnosis of DSM-4 or DSM-5 ADHD based on clinical assessment was required, and participants on short-acting ADHD medications were permitted as long as they refrained from taking medications prior to the visit on the day of testing.

Study design and data acquisition parameters were described in previous manuscripts involving analyses independent from those presented here within the healthy adult (but not ADHD) cohort[14,30,99]. Briefly, participants were scanned in a 3 T Siemens CONNECTOM scanner with 64-channel phased-array head coil at the Athinoula A. Martinos Center for Biomedical Imaging at MGH. We analyzed data from four consecutive fMRI runs that included task performance and experience sampling, and a high-resolution structural MRI scan was used for image co-registration. The T1-weighted structural scan parameters were as follows: TR (2.53 s), TE (1.15 ms), flip angle (7°), TI (1.1 s), resolution (1 mm isotropic). The T2*-weighted functional scan parameters were as follows: multiband acceleration factor (4), TR (1.08 s), TE (30 s), flip angle (60°), slice number (68), resolution (2 mm isotropic).

During task fMRI runs, participants performed the gradCPT[20] with intermittent experience sampling[14]. Each fMRI run lasted ~9 min, depending on the duration of the participants' self-paced ratings during task performance. A continuous stream of visual stimuli, consisting of city and mountain scenes, was presented using Psychophysics Toolbox[100] projection through a mirror from within the scanner bore. Scenes from a set of 10 city and 10 mountain photographs, respectively, gradually transitioned from one to another for the duration of a given run. Scenes were presented randomly, with cities and mountains, respectively, presented pseudorandomly at a rate of ~10%. Using an MRI-compatible button box, participants were instructed press a button with their right index finger each time they saw a city scene but to withhold a press when they saw a mountain scene.

Thought probes appeared after block durations of 44, 52, or 60 s (selected randomly). The number of mountain (target) stimuli presented prior to thought probes was matched across blocks via a biased presentation rate within the 32-s window prior to thought probe onset. In the −12 s before thought probe onset, no targets were presented. In the −12 to −32 s window, the target rate was 8% (outside of the pre-thought probe window, the target rate was 13%). During the thought probe, a question was displayed: "To what degree was your focus just on the task or on something else?" On a self-paced, continuous scale (right anchor: "Only task"; left anchor: "Only else"), participants provided a rating using the middle and ring fingers to move the scale left and right, respectively. They used their thumb to submit their response. Responses were recorded on a graded scale of integers ranging from 0 to 100. Similar to previous work[11], a second question was then presented, which assessed meta-awareness of attentional focus using the question "To what degree were you aware of where your focus was?" Participants provided a self-paced response for this meta-awareness item on a continuous scale recorded on a 0 to 100 graded scale of integers (right anchor: "Aware"; left anchor: "Unaware"). After the participant responded to the second question, the next gradCPT block resumed immediately.

Following completion of the scan session, participants completed an oral interview in which they were asked to what degree their reports of attention off-task during the session was due to (a) external distractions (EDs; e.g., sounds of the scanner), (b) task-related interferences (TRIs), and (c) stimulus-independent thought (SITUT)[5]. For each item, participants provided a rating on a 7-point Likert scale (1 = never; 7 = always)[22]. Subsequently, to confirm the numerical ratings, we asked participants to verbally provide specific examples of the stimulus-independent thoughts and external distractions that had occurred during task performance. Across participants, SITUT ratings were negatively correlated with both ED (Spearman's $\rho = -0.58$, $P = 0.0012$) and TRI ($\rho = -0.47$, $P = 0.011$) ratings, suggesting that it was rare for a participant who reported high SITUT to also report high EDs or TRIs.

*Superstruct study procedures*. We downloaded openly available imaging and phenotypic data from the Brain Genomics Superstruct Project[101] (see, ref. [68] for protocol details). Procedures for this study were approved by the Partners Human Research Institutional Review Board and the Harvard University Committee on the Use of Human Subjects in Research. Briefly, 1570 clinically normal participants (males and females) aged 18–35 completed structural MRI and rs-fMRI. Among these participants, 926 completed at least one run of rs-fMRI and, subsequently, the Daydreaming Frequency Scale[65] (among other various additional behavioral and self-report measures), which was the focus of our analyses of trait SITUT.

Participants were scanned in a Siemens 3 T Tim Trio system with a 12-channel phased-array head coil, either at Harvard University or MGH. During rs-fMRI, participants were instructed to keep their eyes open and to remain awake. The T1-weighted scan parameters were as follows: duration (2 min 12 s), TR (2.2 s), TE (1.5/3.4/5.2/7.0 ms), flip angle (7°), TI (1.1 s), slice number (144), resolution (1.2 mm isotropic). The T2*-weighted scan parameters were as follows: duration (6 min 12 s), TR (3.0 s), TE (30 ms), flip angle (85°), slice number (47), resolution (3.0 mm isotropic).

*Leipzig study procedures.* We downloaded openly available imaging and phenotypic data from the functional connectome phenotyping dataset[102], a component of the MPI-Leipzig Mind-Brain-Body project (see, ref. [103] for protocol details). Procedures for this study were approved by the ethics committee at the medical faculty of the University of Leipzig (097/15-ff). Briefly, 194 healthy participants (males and females) aged 20–75 completed structural MRI and four runs of rs-fMRI. Following each rs-fMRI run, participants completed the Short New York Cognition Questionnaire[104]. In addition, all participants completed the Mind Wandering Deliberate (MW-D) and Mind Wandering Spontaneous (MW-S)[67] scales (among other various additional measures), which was the focus of our analyses of trait SITUT.

Participants were scanned in a 3 T Siemens Magnetom Veria system with a 32-channel head coil at the University of Leipzig. The T1-weighted, 3DMP2RAGE, scan parameters were as follows: duration (8.22 min), TR (5 s), TE (2.92 ms), flip angle 1/2 (4/5°), TI 1/2 (700/2500 ms), slice number (176), resolution (1.0 mm isotropic). The T2*-weighted, gradient-echo echo-planar imaging, scan parameters were as follows for each of the four runs: duration (15 min 30 s), multiband acceleration factor (4), TR (1.4 s), TE (39.4 ms), flip angle (69°), echo spacing (0.67 ms), slice number (64), resolution (2.3 mm isotropic). In the first and third runs, the phase encoding direction was anterior to posterior, whereas in the second and fourth runs, the phase encoding direction was posterior to anterior.

*MIT study procedures.* We analyzed neuroimaging data from a cohort of patients with ADHD who completed neuroimaging at the Athinoula A. Martinos Image Center at the McGovern Institute at MIT. Participants included 59 ADHD adults (males and females) who first underwent a diagnostic interview and completion of questionnaires at MGH and then returned for a neuroimaging session at MIT. All participants underwent a semi-structured interview, the Kiddie Schedule for Affective Disorders and Schizophrenia (K-SADS) to confirm ADHD diagnosis (diagnostic threshold set to 5 for hyperactivity and 6 for inattention symptom counts). All participants also completed the Mind Wandering Questionnaire[66] (among other measures), which was the focus of our analyses of trait SITUT. We documented each patients' current ADHD medication status (data were missing for six participants). Participants using medications were asked to not take their medication for the 48-h period prior to the MRI session. Participants provided written informed consent for procedures approved by the Partners Human Research and MIT institutional review boards.

Inclusion criteria were (1) adults aged 18–45 years; (2) diagnosis of DSM-5 ADHD based on clinical assessment; (3) must understand the nature of the study and verbally provide informed consent. Exclusion criteria were (1) any unstable psychiatric illness (other than ADHD) that, in the judgment of the investigator, may interfere with participation; (2) major sensorimotor handicaps (paralysis, deafness, blindness); (3) currently treated with psychotropic medications other than an ADHD medication; (4) inadequate command of the English language; (5) any contraindications for MRI examination (e.g., metallic implants such as pacemakers, surgical aneurysm clips, or known metal fragments embedded in the body); (6) history of neurological injury or disease; (7) IQ < 80.

Participants completed structural MRI and one rs-fMRI run in a 3 T Siemens Prisma system with a 32-channel head coil. The T1-weighted, MPRAGE, scan parameters were as follows: TR (2.53 s), TE (2.3 ms), flip angle (9°), TI (0.9 s), slice number (176), resolution (1.0 mm isotropic). The T2*-weighted, gradient-echo echo-planar imaging, scan parameters were as follows: duration (7 min), multiband acceleration factor (6), TR (1.0 s), TE (38 ms), flip angle (61°), slice number (66), resolution (2.0 × 2.0 × 2.2 mm isotropic).

**fMRI preprocessing.** We preprocessed each fMRI run individually using identical procedures across all datasets, based on procedures implemented in the CONN toolbox (version 19c (https://www.nitrc.org/projects/conn)[105] and SPM12 in Matlab R2019a (Mathworks Inc., Natick, MA). The pipeline included deletion of the first four volumes, realignment and unwarping[106], and identification of outlier frames (frame-wise displacement >0.9 mm or global BOLD signal change >5 SD)[107]. Functional and anatomical data were normalized into standard MNI space and segmented into gray matter, white matter (WM), and cerebrospinal fluid (CSF) in a unified step[108]. Smoothing of fMRI data was performed using spatial convolution with a Gaussian kernel of 6 mm full-width half-maximum (FWHM).

We next performed fMRI denoising based on linear regression of the following parameters from each voxel: (a) 5 noise components each from minimally-eroded WM and CSF (one-voxel binary erosion of voxels with values above 50% in posterior probability maps), respectively, based on aCompCor procedures[50,109]; (b) 12 motion parameters (3 translation, 3 rotation, and associated first-order derivatives); (c) all outlier frames identified within participants; and d) linear BOLD signal trend within session. In a separate step after nuisance regression[110], data were then temporally filtered with a bandpass of 0.01–0.1 Hz.

Though these specific denoising procedures have been shown to reduce the impact of head motion on functional connectivity[111], excessive head motion can continue to confound estimates[48,49]. We therefore set an absolute threshold such that participants with mean overall frame-wise displacement (FD) of >0.15 mm (based on the Jenkinson method[112]) were excluded from analyses in all datasets. In the Leipzig dataset in which four rs-fMRI runs were obtained within participants, this threshold was based on the mean across runs, and we additionally excluded

individual runs showing an FD value exceeding the 75th percentile plus 1.5 times interquartile range (based on procedures implemented in fsl_motion_outliers)[113]. Within retained participants, we tested for correlations between FD and behavioral outcomes of interest. Also, given that FD can influence observed relationships between functional connectivity and behavior[48], we controlled for FD in analyses focused on relationships between functional connectivity and self-report ratings (see Methods: "Predictive modeling analysis of SITUT in healthy adults").

Notably, ICA-based denoising represents an alternative to aCompCor that can be effective in reducing the impact of head motion on functional connectivity[114]. To further confirm the validity of our predictive model (see Methods: "Predictive modeling analysis of SITUT in healthy adults"), we thus applied an alternative fMRI preprocessing pipeline based on procedures implemented in FSL[113], in line with our previous analysis of the MGH healthy adult sample[14]. Briefly, for each run, this included brain extraction, realignment to middle volume, spatial smoothing with a 5 mm FWHM kernel, application of ICA-AROMA[115], regression of WM and CSF signals, bandpass temporal filtering (0.01–0.1 Hz), and linear registration to T1-weighted and MNI152 space.

**Functional connectivity feature extraction.** Within each dataset, we extracted the preprocessed BOLD time series from the mean across all voxels within each node defined based on previously described intrinsic functional network atlases in MNI space (Shen[116] and Schaefer[117] atlases of 268 whole-brain and 300 cortical regions, respectively). For the MGH experience sampling dataset only, these time series were extracted for each trial based on the ~30-s window (28 TRs) prior to thought probe onset. This window length was chosen based on the task design, which included a matched number of target/non-target stimuli across pre-thought probe windows (see Methods: "MGH study procedures"). Moreover, prior evidence suggests that ~22-40 s functional connectivity estimates can reliably distinguish distinct cognitive tasks[92,93] and intensities of self-reported task-unrelated thoughts[12]. For both the gradCPT experience sampling and rs-fMRI datasets, we also extracted time series across the whole duration of each run. We computed a matrix of functional connectivity values between all region pairs based on the Fisher z-transformed Pearson correlation coefficient of time series.

**Predictive modeling analysis of SITUT in healthy adults.** We adapted the connectome-based predictive modeling (CPM) method[46] to identify functional connectivity patterns predictive of within-participant SITUT. Though CPM has typically been applied to predict individual differences in behavior based on data from whole fMRI runs[40,41,85], our approach is inspired by recent work suggesting that the method can also be sensitive to intra-individual behavioral fluctuations based on shorter time windows within fMRI runs[56]. Our CPM analysis was performed based on the experience sampling and fMRI data from healthy adults (MGH dataset). Though 29 healthy adults participated in the experience sampling study, we retained for this analysis only those who reported (a) trial-by-trial variation in "off-task" ratings (1 participant excluded due to no variation), and (b) a retrospective, post-scan questionnaire rating indicating that "off-task" ratings during thought probes were due to task-unrelated thought that was also stimulus-independent (SITUT; minimum Likert scale rating of 4, as in prior work[23], resulting in exclusion of 11 participants). To further confirm that the retained cohort's "off-task" reports were largely due to SITUT, as opposed to EDs or TRIs, we performed Wilcoxon rank-sum tests (two-sided) comparing SITUT ratings to those of the other two categories.

The 17 included participants (which we term "normative sample") consisted of 9 females and 8 males ($M \pm SD$ age: 25.6 ± 3.6). This sample size is very similar to those used in various previous studies for the purpose of building fMRI-based predictive models that are testable in larger samples[41,43,56,118]. Moreover, given that trial-wise data included 612 samples across and were treated as individual observations in our modeling, the sample size was in line with putative guidelines for prediction analyses[119].

For each 1 of the 17 participants in the normative sample, we generated model-based predictions of trial-wise SITUT ratings based on independent data from all other included participants (i.e., leave-one-participant-out, leave 36 trials out cross-validation). For each cross-validation fold, we computed the correlation between each unique edge (node pair) in the functional connectivity matrix (derived from the Shen or Schaefer atlas) and within-participant z-scored SITUT ratings (i.e., across 576 trials in 16 participants). We then 'masked' the brain-behavior correlations such that only edges correlated with SITUT ratings at the suprathreshold level of $P < 0.01$ (two-tailed) were retained, resulting in positive and negative edge masks (Fig. 1a). For each trial, we computed the dot product between the functional connectivity matrix and each mask and then summed all positive and negative edge values separately. We then calculated a single $S$ (network strength) value, the subtraction of negative edge from positive edge sums. We then fit a linear model, based on all trials in the fold, of the form $MW = \beta*S + c$, where $SITUT$ is SITUT rating. In the held-out-participant, we performed a Pearson correlation between model-predicted SITUT ratings and observed, within-participant normalized SITUT ratings. Importantly, as normalization was performed at the within-participant level, the training and testing data were always kept fully independent[120].

To determine whether predicted versus observed correlations in held-out participants were statistically significant at the group level, we generated a

distribution of null values within each fold. To do so, we repeated all of the same CPM procedures, as described, except the trial assignments of SITUT ratings were randomly permuted (1000 iterations) to obtain null correlation values. At the group level, we performed a Wilcoxon signed-rank test to compare the within-participant prediction (predicted versus observed SITUT) versus the mean of the within-participant null correlation values (significance set at $P < 0.05$, two-tailed). Code for CPM analyses was adapted from publicly available scripts (https://github.com/YaleMRRC/CPM).

We repeated CPM procedures in a series of control analyses, where we performed partial correlations between predicted and observed SITUT within held-out participants for each cross-validation fold. Using these partial correlations, we controlled for the following variables: (1) Head motion, defined as the mean FD value across all 28 frames within each trial; (2) Reaction time (RT) variability during the 30-s pre-thought probe window. We computed RT variability based on the mean of absolute RT variance for behavioral responses to gradCPT city images (i.e., correct commission trials)[14,20]. Behavioral error rates are associated with RT variability within and across individuals[14,20] and could also be indicative of SITUT. However, we did not consider error rates because the majority of 30-s pre-thought probe time windows had 0% error rates (86.1% and 77.9% of windows, respectively, for commission and omission errors across all participants), thus providing insufficient cross-trial variability to enable a powerful analysis; and (3) Sustained Attention and Creativity network strengths (see Methods: "Functional connectivity feature extraction"). For null comparisons (1000 permutations), the random trial assignments of the control variable and SITUT ratings were jointly permuted.

We additionally tested whether there was a relationship between SITUT ratings and RT variability, as in prior work[14], within the 17-participant healthy sample analyzed here. We performed a linear mixed-effects model analysis with participants entered as random effect, within-participant normalized SITUT as the dependent variable, and within-participant normalized RT variability as fixed effect. We then performed $F$ tests on the model coefficients, with significance set at $P < 0.05$ (Satterthwaite's approximation.

To facilitate comparison with prior work[40,41,56], our main subsequent analyses focus on the model parameters obtained from these procedures (which we term the "SITUT-CPM") based on the Shen268 atlas (unless stated otherwise). For external validation analyses assessing generalizability in other datasets, as well as analyses of edge network identities (see Methods: "Analysis of functional neuroanatomical patterns contributing to the SITUT-CPM"), we computed SITUT-CPM parameters and positive and negative masks based on data from all participants in the normative sample (i.e., a single fold). The final Shen268 positive and negative edge masks, respectively, contained an average of 72.7 and 73.8% network overlap with the distinct masks obtained in the 17 individual folds from the internal validation analysis.

**Comparison of the SITUT-CPM with previously published CPMs**. Within the MGH dataset of healthy participants, we performed comparisons between the SITUT-CPM and two other CPMs previously reported in the literature, the Sustained Attention[41] and Creativity[40] CPMs (SA-CPM and Cr-CPM), derived from the Shen268 atlas. We computed the proportion of SITUT-CPM positive and negative mask edges (out of total edges within each mask) that showed overlap with the positive and negative masks for these two published CPMs. To determine the significance of network overlap based on the hypergeometric cumulative density function (as done elsewhere[121]) implemented in Matlab as $P = 1\text{-}hygecdf(x, M, K, n)$ where $x$ was the number of overlapping edges between two CPMs, $M$ was the total number of edges (including those within or not within the masks), $K$ was the number of mask edges in the SITUT-CPM, and $n$ was the number of mask edges in the other CPM (SA-CPM or Cr-CPM).

In addition, we computed "network strength" of the SA-CPM and Cr-CPM within our data, based on the dot product between trial-wise functional connectivity matrices and the positive and negative masks of these CPMs. The network strength values were defined as the subtraction of negative edge from positive edge sums. We used the same procedures to define SITUT-CPM network strength based on positive and negative SITUT-CPM masks derived from the entire dataset of 17 participants in the normative sample. We performed within-participant, trial-wise correlations between SITUT-CPM network strength versus sustained attention and creativity network strengths. At the group level, we statistically tested whether within-participant Fisher z-transformed correlations were significantly different from zero using a Wilcoxon signed rank (significance set at $P < 0.05$, two-tailed).

Finally, we tested whether SA-CPM was significantly predictive of trial-wise SITUT ratings within our normative sample. To do so, we correlated observed trial-wise SITUT ratings with predicted sustained attention from the SA-CPM within each participant. Across participants, we performed significance testing using the same procedures as those described in Methods: "Functional connectivity feature extraction". At the group level, we performed a Wilcoxon signed-rank test to compare the within-participant prediction (predicted versus observed SITUT) versus the mean of the within-participant null correlation values (significance set at $P < 0.05$, two-tailed).

**Analysis of functional neuroanatomical patterns contributing to the SITUT-CPM**. We visualized the edges comprising SITUT-CPM masks (Fig. 1d) using the

BioImage Suite Connectivity Visualization Tool (https://bioimagesuiteweb.github.io/webapp/connviewer.html). To gain insight into the neuroanatomical patterns that contributed to the SITUT-CPM within the normative group of healthy adults (MGH sample), we assigned each node to one of seven or 17 canonical Yeo-Krienen[58] intrinsic functional networks. For these analyses, we used the Schaefer atlas of 300 cortical regions, which includes a Yeo-Krienen network label for each node[117]. To describe our results, we use the labels provided with the publicly available Schaefer atlas (https://github.com/ThomasYeoLab/CBIG/tree/master/stable_projects/brain_parcellation/Schaefer2018_LocalGlobal) with an exception made for FPCN subnetworks found in the 17-network atlas; we flipped the provided FPCN$_A$ and FPCN$_B$ subnetwork labels so that our descriptions are consistent with prior work[62] in which the subnetwork including rostrolateral prefrontal cortex was termed FPCN$_A$.

Based on the Schaefer atlas, the SITUT-CPM included 385 and 155 edges, respectively, in the positive and negative masks. We assigned each of these edges to one of 28 within- or between-network Yeo-Krienan pairs. For positive and negative masks, we identified the top five network pairs showing the largest number of edges contributing to the masks. For illustration and the purpose of interpretation (Fig. 2b, d), we extracted the mean functional connectivity of each of these network pairs during the top and bottom 50% of trials with the highest and lowest SITUT ratings, respectively.

In addition, in a complementary analysis to the CPM approach, we performed a univariate analysis of the relationship between SITUT rating and within-/between-network connectivity. For each of 28 network pairs, we computed the mean functional connectivity across all edges that were assigned to a given network pair within each trial. We then performed a linear mixed-effects model analysis with participant entered as random effect, within-participant normalized SITUT as the dependent variable, and mean network-pair functional connectivity as fixed effect. We performed $F$ tests on the model coefficients, with significance set at $P < 0.05$ (Satterthwaite's approximation, two-tailed, false-discovery rate-corrected for all 28 network pairs). In addition, within each participant, we computed the trial-wise correlation between SITUT-CPM network strength and network-pair functional connectivity.

Given that our comparisons between the CPM and univariate analyses suggested a potentially selective, key role of DMN-FPCN edges, we repeated the CPM analysis procedures (see Methods: "Predictive modeling analysis of SITUT in healthy adults") using DMN-FPCN edges (Schaefer atlas) only. We also performed an alternative version of the CPM procedures with all DMN-FPCN edges selectively deleted from training and testing data (i.e., virtually lesioned) and with all other Schaefer atlas edges retained.

**Within-participant SITUT prediction in ADHD**. We tested generalizability of the SITUT-CPM within a cohort distinct from that used to derive the SITUT-CPM, namely 20 adults diagnosed with ADHD ($M \pm SD$ age: 26.2 ± 8.5; 10 females, 10 males) who underwent the same task performance and experience sampling procedures with the same MRI scanner (MGH sample). Post-scan questionnaire ratings indicated that "off-task" ratings in the ADHD group were largely due to stimulus-independent thought ($M \pm SD$ rating: 5.0 ± 1.5 on 7-point Likert scale), confirming that participants with ADHD engaged in SITUTs. For this analysis, each patient with ADHD and their 36 trials were treated as a held-out participant to test SITUT-CPM predictions (based on the model generated within healthy adults).

We computed SITUT-CPM network strength on a trial-by-trial basis within each patient, and then applied the linear model from the SITUT-CPM to generate predicted SITUT ratings. We then calculated the Pearson correlation coefficient for predictive versus observed SITUT ratings. To generate null comparison values, we repeated all of the same procedures for each held-out ADHD participant, except trial assignments of SITUT ratings were randomly permuted across all participants with ADHD (1000 iterations) to obtain null correlation values. At the group level, we performed a Wilcoxon signed-rank test to compare the ADHD within-participant prediction (predicted versus observed SITUT) versus the mean of the within-participant null correlation values (significance set at $P < 0.05$, two-tailed). Moreover, to further assess the generalizability between ADHD and healthy adult datasets, we used the full ADHD dataset to generate CPM parameters (linear model and positive and negative mask edges) and tested this model within held-out healthy adult participants (using identical procedures those described in this section and in Methods: "Predictive modeling analysis of SITUT in healthy adults".

**Comparison of SITUT-CPM network strength in healthy adults versus adults with ADHD during task performance**. In addition to testing whether the SITUT-CPM within-participant predictions were generalizable across healthy individuals and individuals with ADHD, we tested whether these two groups reported different levels of SITUT during experience sampling and showed differences in SITUT-CPM network strength during task performance. We included 28 healthy adults (after excluding one participant excluded due to no variation in "off-task" ratings; 13 males, 15 females; $M \pm SD$ age: 26.2 ± 3.8) and all 20 participants with ADHD in these analyses. At the behavioral level, we compared groups in terms of mean ratings across trials (computed within each participants for (a) off-task ratings, and (b) meta-awareness ratings. At the neural level, we compared groups in terms of mean SITUT-CPM network strength (averaged across runs). We performed

statistical comparisons between groups using Wilcoxon rank-sum tests (significance set at $P < 0.05$, two-tailed). The healthy and ADHD groups showed no significant difference with one another in mean frame-wise displacement across runs ($P = 0.31$, Wilcoxon rank-sum test).

**Prediction of trait SITUT from rs-fMRI in healthy adults**. In the Superstruct and Leipzig datasets, we tested whether SITUT-CPM predictions generalize to predictions of trait SITUT from rs-fMRI data. In the Superstruct dataset, we excluded 15 out of 926 total participants due to excessive head motion, resulting in 911 participants included. In the Leipzig dataset, we excluded 44 out of 188 total participants due to excessive head motion, resulting in 144 total participants included, and we computed the mean functional connectivity within each participant across all included rs-fMRI runs. The higher relative levels of head motion in the Leipzig dataset may have been due to the longer total scan and individual run durations.

For each participant, we computed SITUT-CPM network strength based on the dot product of the resting-state functional connectivity matrix and the SITUT-CPM positive and negative edge masks, and we computed predicted SITUT scores based on the SITUT-CPM linear model parameters. We then computed the Spearman's rank correlation coefficient between predicted SITUT scores and observed trait SITUT scores (Daydreaming Frequency Scale in Superstruct dataset; Spontaneous and Deliberate Mind Wandering score, each separately, in Leipzig dataset) (significance set at $P < 0.05$, two-tailed). Among Superstruct participants included in analyses, there was no significant correlation between frame-wise displacement and Daydreaming Frequency Scale score (Spearman's $\rho = -0.031$, $P = 0.35$).

Among Leipzig participants included in analyses, frame-wise displacement was significantly negatively correlated with Spontaneous (Spearman's $\rho = -0.17$, $P = 0.038$) and Deliberate Mind Wandering (Spearman's $\rho = -0.20$, $P = 0.014$). Thus, to account for head motion and other potentially confounding factors, we repeated correlation analyses between SITUT-CPM predictions and behavior using partial Spearman correlations controlling for individual differences in (a) mean frame-wise displacement (in both datasets); and (b) age (based on 5-year bins[103], in the Leipzig dataset).

To assess specificity of model prediction in the Superstruct dataset, we computed the Spearman's $\rho$ value for 66 total behavioral and self-report measures that were obtained in addition to daydreaming frequency, such as performance on Flanker and Mental Rotation tasks, IQ, and self-report measures of personality, depression, and anxiety (see[68] for full description; depending on data availability, these correlations were performed with sample sizes ranging from 892 to 911 participants). Finally, as a further test of specificity, we repeated analyses (Superstruct and Leipzig) using the SA-CPM to compare predicted sustained attention ability versus observed trait SITUT.

**Prediction of trait SITUT from rs-fMRI in ADHD**. In the ADHD rs-fMRI dataset (collected at MIT), we tested whether SITUT-CPM network strength was different between subgroups of patients with high versus low trait SITUT, based on previously established clinical relevance of this subgrouping[8]. We excluded 10 out of 59 total participants due to excessive head motion, resulting in 49 participants included. Using previously defined criteria[8], we then classified these participants into those with high and low trait SITUT (MWQ score >23 and <24, respectively). We compared the two groups in terms of head motion and age (Wilcoxon rank-sum test, two-sided) as well as sex (F/M ratio) and current ADHD medication status (proportion of patients using medications). Data were missing on ADHD medication status for 1 participant in the high SITUT and 2 participants in the low SITUT subgroups.

We used the SITUT-CPM parameters to compute network strength within each participant, and then we compared network strengths between subgroups using a Wilcoxon rank-sum test (significance set at $P < 0.05$, two-tailed). In addition, we performed a complementary analysis where MWQ score was treated as a continuous variable, and we computed SITUT-CPM-predicted versus observed MWQ scores (Spearman correlation, including partial correlations controlling for head motion and age). To gain insight into the clinical relevance of MWQ scores within our ADHD cohort, we performed correlations (Spearman) between MWQ and K-SADS symptom severity scores for inattention and hyperactivity (including the diagnostic scores and the current scores based on the past 6 months).

**Intra-individual analysis of SITUT-CPM network dynamics during rs-fMRI**. In the Leipzig dataset, we performed analyses of the temporal dynamics of subjective experience and SITUT-CPM network strength across four consecutive rs-fMRI runs. On the SNYCQ, which participants completed after each run, we focused on the item "my thoughts involved my surroundings" as an indicator of fluctuations in stimulus-independence of experience across runs. We additionally analyzed the item "I was fully awake" as an indicator of subjective wakefulness. In these analyses, we excluded 3 participants who reported 0 on at least one run for the wakefulness item. After further excluding runs due to excessive head motion, the following numbers of participants were included: run 1 ($n = 140$) run 2 ($n = 165$), run 3 ($n = 140$), and run 4 ($n = 146$).

We computed SITUT-CPM network strength based on the dot product between the within-run functional connectivity matrix and the SITUT-CPM positive and negative masks. For SNYCQ item ratings and network strength, we performed linear mixed-effects model analyses with participants entered as random effect, rating (or network strength) as a dependent variable, and run number as fixed effect. We performed $F$ tests on the model coefficients, with significance set at $P < 0.05$ (Satterthwaite's approximation, two-tailed). Using post hoc Wilcoxon rank-sum tests, we performed statistical comparisons between each pair of runs, with significance set at $P < 0.05$ (false-discovery-rate corrected, two-sided). In addition, to test for a relationship between SITUT-CPM network strength and subjective experience across runs, we performed a linear mixed-effects model analysis with participant as random effect, "my thoughts involved my surroundings" rating as dependent variable, and network strength as fixed effect.

**Reporting summary**. Further information on research design is available in the Nature Research Reporting Summary linked to this article.

## Data availability

Data from the Brain Genomics Superstruct (https://www.neuroinfo.org/gsp) (https://doi.org/10.7910/DVN/25833) and Leipzig (https://www.nitrc.org/projects/mpilmbb) (https://doi.org/10.18112/openneuro.ds000221.v1.0.0) datasets are publicly available. For the MGH and MIT datasets, original functional and structural MR images can be made available upon reasonable request to the authors with mandatory ethics approval and required data use agreements with Massachusetts General Hospital and/or Massachusetts Institutes of Technology. Source data are provided with this paper.

## Code availability

Code for CPM analyses is shared along with visualization tools (Fig. 2) and two versions of SITUT-CPM model parameters (based, respectively, on Shen268 and Schaefer300 atlases) at https://github.com/swglab/CPM_CONN[122].

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

## Acknowledgements

We thank Alfonso Nieto Castañón for technical assistance and Clay Riley, Yuwen Hung, Sean Dea Houlihan, Schuyler Gaillard, Caroline Kelberman, and Elizabeth Noyes for assistance with study coordination and data collection. We thank Monica Rosenberg and Roger Beaty for sharing the Sustained Attention and Creativity connectome-based predictive models, respectively. We thank Daniel Margulies and Mark Lauckner for helpful discussions regarding interpretation of the Leipzig data. We thank Jonathan Smallwood for helpful comments on a draft of the manuscript. The MGH study was supported by grants from National Institute of Health R01 HD067744-01A1 (to E.M.V.) and the Athinoula A. Martinos Center for Biomedical Imaging (NCRR P41RR14075 & P41 EB015896). The MIT study was supported by the Poitras Center for Psychiatric Disorders Research at the McGovern Institute for Brain Research. This work was completed in part using the Discovery cluster, supported by Northeastern University's Research Computing team.

## Author contributions

A.K.: Conceptualization, methodology, software, formal analysis, data curation, investigation, visualization, writing–original draft, writing—review and editing. M.E.: Conceptualization, methodology, software, writing—review and editing. J.C.: Investigation, data curation. A.G.: Investigation, data curation, writing—review and editing. M.U.:

Conceptualization, Methodology, Writing—review and editing. J.B.: Funding acquisition, project administration, conceptualization, methodology, writing—review and editing. J. D.E.G.: Supervision, funding acquisition, project administration, conceptualization, methodology, writing—review and editing. E.M.V.: Supervision, funding acquisition, project administration, conceptualization, methodology, writing—review and editing. S. W.-G.: Supervision, funding acquisition, project administration, conceptualization, methodology, writing—review and editing.

## Competing interests

A.K., M.E., E.M.V., J.C., A.G., M.U., J.D.E.G., and S.W.G. have no competing interests. Dr. Joseph Biederman is currently receiving research support from the following sources: AACAP, Feinstein Institute for Medical Research, Food & Drug Administration, Genentech, Headspace Inc., NIDA, Pfizer Pharmaceuticals, Roche TCRC Inc., Sunovion Pharmaceuticals Inc., Takeda/Shire Pharmaceuticals Inc., Tris, and NIH. He receives honoraria from the MGH Psychiatry Academy for tuition-funded CME courses. Dr. Biederman's program has received departmental royalties from a copyrighted rating scale used for ADHD diagnoses, paid by Biomarin, Bracket Global, Cogstate, Ingenix, Medavent Prophase, Shire, Sunovion, and Theravance; these royalties were paid to the Department of Psychiatry at MGH. In 2020: Through MGH corporate licensing, Dr. Biederman has a US Patent (#14/027,676) for a non-stimulant treatment for ADHD, a US Patent (#10,245,271 B2) on treatment of impaired cognitive flexibility, and a patent pending (#61/233,686) on a method to prevent stimulant abuse. In 2019, Dr. Biederman was a consultant for Akili, Avekshan, Jazz Pharma, and Shire/Takeda. He received research support from Lundbeck AS and Neurocentria Inc. Through MGH CTNI, he participated in a scientific advisory board for Supernus. In 2018, Dr. Biederman was a consultant for Akili and Shire. In 2017, Dr. Biederman received research support from the Department of Defense and PamLab. He was a consultant for Aevi Genomics, Akili, Guidepoint, Ironshore, Medgenics, and Piper Jaffray. He was on the scientific advisory board for Alcobra and Shire.
