## [Peer Review File · Nature Communications]

Reviewers' Comments:

Reviewer #1:

Remarks to the Author:

Thank you for inviting me to review this paper by Kucyi and colleagues. The authors tackle an important issue in mind wandering research: whether mind wandering can be linked to a generalisable network pattern, despite different behavioural assays used to probe it. This feels like a timely investigation. The mind wandering field is currently occupied with nuanced definitions of how to operationalise mind wandering across psychological tasks, so it is encouraging to see these results establishing network correlates that generalise across individuals, across settings, and across different behavioural assays of mind wandering.

The authors used connectome-based predictive modelling (CPM) to model and predict trial-wise, intra-individual fluctuations in stimulus-independent, task-unrelated thought (SITUT). Regions across the default and frontoparietal networks were identified by the CPM. These were then extensively validated in a clinical (ADHD) sample, and in three separate large resting state samples, providing convergence across state (task-based) and trait (resting-state) assessments of mind wandering.

I think this paper is making an important contribution by identifying neural correlates of mind wandering that appear to be robust and generalisable. The manuscript was an enjoyable read – nicely written and easy to follow.

I have the following comments that I hope are useful to the authors:

- The authors strongly emphasise potential clinical implications of their work, based on the CPM validation they show in two ADHD cohorts. For me, a couple of things jumped out that seemed relevant to the potential clinical interpretation of the findings:

- 1) The authors cite evidence that high levels of mind wandering in ADHD are associated with worse clinical features. Is this something that bears out in their data, i.e., if their mind wandering measures are associated with clinical severity this could be worth reporting.
- 2) Related to the above point, dividing the MIT ADHD cohort into high vs. low mind wandering to compare CPM network strengths make sense. It would be helpful to know whether these groups were comparable (or not) on relevant clinical measures that we could imagine might impact network connectivity (i.e., age, clinical severity, medication use).

I appreciate this is not a clinical paper and there is no need for it to be derailed by an extensive focus on the clinical cohorts. But given the emphasis the authors place on the potential clinical relevance of their findings, there seemed scope for a bit more nuance in reporting the clinical results.

- For the resting state external validations, in all three, frame-wise head motion is controlled for in the correlation. It looks like age is only controlled for in the Leipzig dataset? I see the Leipzig dataset has a much larger age range, but if age is a relevant factor then a range of 18-45 in the MIT ADHD dataset might not be considered trivial. I wondered why age wasn't considered in the other datasets as well?

- I'm not directly familiar with the gradCPT task, so perhaps I'm missing something obvious. But I couldn't see why RT variability would be computed only based on correct behavioural responses to gradCPT city images. I would have thought incorrect responses would be most relevant and most representative of periods of SITUT?

- This may be off base and the authors can feel free to disregard. But I noticed in Figure 4b, among those measures showing the strongest anti-correlation with the SITUT-CPM prediction look to be measures from the Flanker task. This seems to complement the finding that SITUT-CPM was

negatively correlated with the Sustained Attention CPM, perhaps further supporting the independence of the SITUT-CPM from networks related to attention. Not sure the authors agree this is worth highlighting, just struck me as a potentially cool extra finding.

Minor:

- Table 1. Would be helpful to have the questionnaire abbreviations in a legend below the table.

Reviewer #2:

Remarks to the Author:

Overview:

Across a series of studies, Kucyi and colleagues develop and validate a brain-network marker of stimulus-independent task-unrelated thinking (SITUT) – an important source of within and between-subject variability in behavior. The authors develop their whole-brain predictive marker using connectome-based predictive modelling on functional connectivity patterns while participants complete the gradCPT task. Self-report questions and behavioral performance identify trials in which SITUTs occur, and to what degree. The authors subsequently apply their model to independent groups of participants (including those with and without ADHD), and across different environmental and task contexts (different MRI scanners, during the “resting” state).

Aside from providing the field with more basic science insight into the functionally-connected regions that support SITUT, the authors’ findings also have important relevance for clinical populations struggling from an over-abundance or an under-abundance of SITUT. Overall, I was particularly impressed with this manuscript, and feel that its significance, methodology, scope and communication are appropriate for Nature Communications. I do have a few comments / concerns, however, which I outline below.

Moderate/Major Comments:

1) My main concern is that it seems to me that the SITUT marker may be somewhat dependent on the nature of the ongoing task. In particular, the connectome-based predictive marker of SITUT was developed while participants completed a gradCPT task involving detecting manmade vs. natural scenes. However, scene detection involves medial temporal and retrosplenial regions that partially overlap with the default mode network (DMN). Considering that connectivity of these regions may aid in performance of the ongoing task, the additional involvement of these “task-related regions” in SITUT may end up being masked by their role in the gradCPT task. Although the authors apply their SITUT marker to periods of rest, the effects are not extremely strong, perhaps because some regions involved in SITUT may not appear in the marker. I would like to see the authors raise this possible issue in the discussion. Along these lines, I would like to see a limitations section in the manuscript.

2) The authors assess “task-unrelated thought” throughout the gradCPT task, but “external distractions” are assessed only retrospectively at the end of the task. Participants who endorsed external distractions during this retrospective assessment were then excluded from analyses, leaving a sample size of N=17. I think it would be helpful if the authors were to expand on their procedures surrounding this topic, including how many people were excluded from analysis and what rating of external distractions was considered acceptable versus unacceptable enough to warrant inclusion or exclusion. I would also like to see the authors raise the retrospective assessment of external distractions as a study limitation. In particular, participants may not accurately remember what proportion of their stimulus independent thoughts were internal versus external, and other experience sampling studies suggest that stimulus-dependent distractions are somewhat frequent (e.g. ~25% of probes, Stawarczyk et al., Neural Correlates of Ongoing Conscious Experience: Both Task-Unrelatedness and Stimulus-Independence Are Related to Default Network Activity, PLOS One, 2011).

3) I think it would be informative if the authors were to consider pitting the SITUT-CPM map and the

Sustained Attention-CPM map against each other in the service of examining which map is a better predictor of ADHD presence and symptomatology. The authors could also consider estimating the combined predictive validity from the two maps.

4) It strikes me that the healthy participants might be more likely to deliberately engage in SITUT during the gradCPT task than participants with ADHD (who may do so more spontaneously). If this is the case, I would expect FPCN-DMN connectivity to be stronger in healthy adults if such individuals are deliberately holding their attention on SITUTs. Perhaps the authors could speculate on such a possible source of observed differences in SITUT-CPM between groups? Relatedly, DMN-FPCN connectivity may be higher in healthy individuals if DMN-FPCN connectivity is related to awareness. The authors indicated that they did not analyze the question pertaining to awareness, but doing so may help shed light on the implications of the authors' findings. Overall, these two hypotheses seem to fit with the authors' findings that the DMN-FPCN connectivity patterns only significantly predict SITUT in healthy individuals.

Minor Comments:

5) It would be helpful if the authors were to provide more clarity on their sources for the terms "DMN-A", "DMN-B", "DMN-C", "FPCN-A" and "FPCN-B". Citations would be helpful here, particularly because the terms "DMN-A" and "DMN-B" have also been used to describe a 2-solution DMN parcellation (see Buckner & DiNicola, "The brain's default network: updated anatomy, physiology and evolving insights." *Nature Reviews Neuroscience*, 2019). Do the terms FPCN-A and FPCN-B come from Dixon et al., "Heterogeneity within the frontoparietal control network and its relationship to the default and dorsal attention networks" *PNAS*, 2018?

6) On page 9, the following sentence is a bit confusing. "In summary, although SITUT-CPM predictions were based on a complex, distributed pattern of interacting networks, key components associated with increased SITUT were a) decreased DMNA-FPCNA anticorrelation; and b) increased DMNA-SMNA correlation (Fig. 2g)." Unless I'm misunderstanding, did the authors mean to write "decreased" DMNA-SMNA correlation, rather than "increased" correlation?

7) In figure 1d, the transparent lateral brain makes it difficult to visually distinguish between medial versus lateral views, which makes interpretation of the individual regional connectivity patterns a bit challenging. Do the authors have a clearer way to plot lateral and medial views separately?

8) Figure 2g was initially somewhat confusing to me, particularly because I found myself trying to link the background colors "red/pink" to positive predictive links, and "blue" to negative predictive links. Perhaps the authors could choose a different background color scheme.

Signed: Jessica Andrews-Hanna

Response to Reviewers: NCOMMS-20-33465

Reviewer #1

Thank you for inviting me to review this paper by Kucyi and colleagues. The authors tackle an important issue in mind wandering research: whether mind wandering can be linked to a generalisable network pattern, despite different behavioural assays used to probe it. This feels like a timely investigation. The mind wandering field is currently occupied with nuanced definitions of how to operationalise mind wandering across psychological tasks, so it is encouraging to see these results establishing network correlates that generalise across individuals, across settings, and across different behavioural assays of mind wandering.

The authors used connectome-based predictive modelling (CPM) to model and predict trial-wise, intra-individual fluctuations in stimulus-independent, task-unrelated thought (SITUT). Regions across the default and frontoparietal networks were identified by the CPM. These were then extensively validated in a clinical (ADHD) sample, and in three separate large resting state samples, providing convergence across state (task-based) and trait (resting-state) assessments of mind wandering.

I think this paper is making an important contribution by identifying neural correlates of mind wandering that appear to be robust and generalisable. The manuscript was an enjoyable read – nicely written and easy to follow.

>>> **RESPONSE:** We appreciate the reviewer's recognition of the timeliness and importance of our manuscript as well as the robustness of our findings.

I have the following comments that I hope are useful to the authors:

- The authors strongly emphasise potential clinical implications of their work, based on the CPM validation they show in two ADHD cohorts. For me, a couple of things jumped out that seemed relevant to the potential clinical interpretation of the findings:

1) The authors cite evidence that high levels of mind wandering in ADHD are associated with worse clinical features. Is this something that bears out in their data, i.e., if their mind wandering measures are associated with clinical severity this could be worth reporting.

>>> **RESPONSE:** We agree that it would be informative to link SITUT tendency with ADHD clinical features within our dataset. As such, we have now examined the relationship between MWQ score and severity of inattention and hyperactivity symptoms, as assessed with Kiddie Schedule for Affective Disorders and Schizophrenia (K-SADS) semi-structured interview. We have updated the

Methods to describe these analyses, and we now report the following in the Results (Section 3.3):

“...across the entire group, MWQ score was positively associated with inattention and hyperactivity symptom severity within the past 6 months (inattention: $\rho = 0.34$, $P = 0.017$; hyperactivity: $\rho = 0.34$, $P = 0.017$) and showed a weaker, but also positive association with diagnostic symptom severity (inattention: $\rho = 0.23$, $P = 0.11$; hyperactivity: $\rho = 0.23$, $P = 0.12$), in line with prior findings from a larger cohort of ADHD patients.¹”

As indicated, the relationship between MWQ and clinical features does bear out, to some degree, within our dataset (though we acknowledge that our sample size was more limited than that of the cited study, which did not include a neuroimaging component¹). We note that that the ρ and P values above appear identical for the 6-month inattention versus hyperactivity correlations, and we confirmed that this is not an error (the values diverge at 3 decimal places, but we report 2 decimal places for consistency with the rest of the manuscript).

2) Related to the above point, dividing the MIT ADHD cohort into high vs. low mind wandering to compare CPM network strengths make sense. It would be helpful to know whether these groups were comparable (or not) on relevant clinical measures that we could imagine might impact network connectivity (i.e., age, clinical severity, medication use).

I appreciate this is not a clinical paper and there is no need for it to be derailed by an extensive focus on the clinical cohorts. But given the emphasis the authors place on the potential clinical relevance of their findings, there seemed scope for a bit more nuance in reporting the clinical results.

>>> **RESPONSE:** We appreciate this suggestion, and we have now expanded our analyses of factors that may be compared across the high and low SITUT groups. In our original manuscript, we had reported comparisons of age, head motion, and sex in the Methods. For better clarity and flow, we have now moved those findings to the Results section. Additionally, we have now compared the two groups in terms of current medication status. We have updated the Methods section appropriately and have reported the following in the Results (Section 3.3):

“These two groups did not show significant differences in head motion ($P = 0.43$, Wilcoxon rank sum test, $M \pm SD$ of 0.092 ± 0.027 mm for high SITUT participants, 0.088 ± 0.027 mm for low SITUT participants), age ($P = 0.097$; $M \pm SD$ of 32.0 ± 7.5 for high SITUT participants, 28.2 ± 7.0 for low SITUT participants) sex (high SITUT: 18 females, 13 males; low SITUT: 7 females, 11 males), or ADHD medication status (high SITUT: 27.6% using medication; low SITUT: 25% using medication).”

Our response to the reviewer's previous comment (point #1) also addresses the relationship between MWQ and clinical severity. In summary, the two groups appeared to be matched in terms of age, head motion, sex and current medication use, but high compared to low SITUT individuals were more likely to exhibit increased severity of clinical symptoms (both inattention and hyperactivity). We feel that these new analyses add significant clarity that will help readers to better interpret the potential clinical relevance of our findings.

- For the resting state external validations, in all three, frame-wise head motion is controlled for in the correlation. It looks like age is only controlled for in the Leipzig dataset? I see the Leipzig dataset has a much larger age range, but if age is a relevant factor then a range of 18-45 in the MIT ADHD dataset might not be considered trivial. I wondered why age wasn't considered in the other datasets as well?

>>> **RESPONSE:** We originally controlled for age only in the Leipzig dataset due to the particularly large age range in that dataset. However, we agree with the reviewer that it is worthwhile to consider age as a potential factor in the other resting state external validations, even though the age ranges were smaller. As such, we have now controlled for age in the other two resting state datasets (Superstruct and MIT ADHD) and have reported the partial correlation values in the manuscript. As expected, age did not significantly affect SITUT-CPM predictions in those two datasets (the changes in correlations were near zero).

- I'm not directly familiar with the gradCPT task, so perhaps I'm missing something obvious. But I couldn't see why RT variability would be computed only based on correct behavioural responses to gradCPT city images. I would have thought incorrect responses would be most relevant and most representative of periods of SITUT?

>>> **RESPONSE:** We appreciate that the rate of incorrect behavioral responses, in addition to RT variability for correct responses, could be associated with SITUT. Prior work, including our own, indicates that behavioral error rates are associated with RT variability within and across individuals,^{2,3} suggesting that these two metrics may share overlapping variance. However, given the relative ease of the gradCPT in our study (we used a slow version with ~1.3 sec inter-stimulus intervals to enable variation in SITUT), and the limited total number of stimuli presented within each of the 30-second pre-thought probe windows, the error rates were low (0% for the majority of windows for both errors of commission and errors of omission). Thus there was insufficient variability in error rates across trials to enable a powerful analysis. We now note this issue in the Methods (Section 5) as follows:

“Behavioral error rates are associated with RT variability within and across individuals^{2,3} and could also be indicative of SITUT. However, we did not consider error rates because the majority of 30-second pre-thought probe time

windows had 0% error rates (86.1% and 77.9% of windows, respectively, for commission and omission errors across all participants), thus providing insufficient cross-trial variability to enable a powerful analysis.”

Additionally, we have now further highlighted that RT variability (for correct trials) is consistently identified as a correlate of SITUT throughout the literature. We have added the following text (including relevant references) to the Results section:

“...reaction time (RT) variability, indicative of sustained attention,²⁻⁴ and previously shown to be correlated with SITUT.^{3,5-8}”

- This may be off base and the authors can feel free to disregard. But I noticed in Figure 4b, among those measures showing the strongest anti-correlation with the SITUT-CPM prediction look to be measures from the Flanker task. This seems to complement the finding that SITUT-CPM was negatively correlated with the Sustained Attention CPM, perhaps further supporting the independence of the SITUT-CPM from networks related to attention. Not sure the authors agree this is worth highlighting, just struck me as a potentially cool extra finding.

>>> **RESPONSE:** Indeed, this is an interesting aspect of our findings that further highlights the specificity of SITUT-CPM predictions. As noted by the reviewer, the trends were generally toward negative correlations for the Flanker test measures (in line with the negative relationship with the Sustained Attention CPM); however, such correlations were very weak or near zero. Overall, we agree that the findings regarding associations with Flanker test performance are worth highlighting, and we have now modified the relevant Results text as follows:

“Among this entire set of individual outcomes, SITUT-CPM prediction showed the strongest correlation with DDFS score, including in comparison to correlation with performance on a Flanker task purported to more generally capture attentional and inhibitory control (Fig. 4b).”

Minor:

- Table 1. Would be helpful to have the questionnaire abbreviations in a legend below the table.

>>> **RESPONSE:** We thank the reviewer for this suggestion and have now included a legend with Table 1.

Reviewer #2

Overview:

Across a series of studies, Kucyi and colleagues develop and validate a brain-network marker of stimulus-independent task-unrelated thinking (SITUT) – an important source of within and between-subject variability in behavior. The authors develop their whole-brain predictive marker using connectome-based predictive modelling on functional connectivity patterns while participants complete the gradCPT task. Self-report questions and behavioral performance identify trials in which SITUTs occur, and to what degree. The authors subsequently apply their model to independent groups of participants (including those with and without ADHD), and across different environmental and task contexts (different MRI scanners, during the “resting” state).

Aside from providing the field with more basic science insight into the functionally-connected regions that support SITUT, the authors’ findings also have important relevance for clinical populations struggling from an over-abundance or an under-abundance of SITUT. Overall, I was particularly impressed with this manuscript, and feel that its significance, methodology, scope and communication are appropriate for Nature Communications. I do have a few comments / concerns, however, which I outline below.

>>> **RESPONSE:** We are glad that the reviewer was impressed with our manuscript, and we thank them for noting the clinical relevance and significance of our work.

Moderate/Major Comments:

1) My main concern is that it seems to me that the SITUT marker may be somewhat dependent on the nature of the ongoing task. In particular, the connectome-based predictive marker of SITUT was developed while participants completed a gradCPT task involving detecting manmade vs. natural scenes. However, scene detection involves medial temporal and retrosplenial regions that partially overlap with the default mode network (DMN). Considering that connectivity of these regions may aid in performance of the ongoing task, the additional involvement of these “task-related regions” in SITUT may end up being masked by their role in the gradCPT task. Although the authors apply their SITUT marker to periods of rest, the effects are not extremely strong, perhaps because some regions involved in SITUT may not appear in the marker. I would like to see the authors raise this possible issue in the discussion. Along these lines, I would like to see a limitations section in the manuscript.

>>> **RESPONSE:** The reviewer raises a key issue regarding the task context in which we defined the SITUT-CPM. We agree that it is possible that our resting state-based predictions, though robust across three datasets, were weakened due to potential ‘masking’ effects (e.g. in DMN) noted by the reviewer. There are also other key limitations of our approach to defining the CPM, such as our

reliance on 30-second windows that may not directly correspond to the time course of individual episodes of SITUT. As suggested, we have now included a Limitations sections in the Discussion where we discuss these issues as follows:

“A key limitation concerns the specific task context (gradCPT) in which we defined the SITUT-CPM. During the gradCPT, visual detection of scenes (mountains and cities) engages brain regions (e.g. medial temporal and retrosplenial^{2,9}) that could overlap, or dynamically interact, with key networks involved in SITUT (e.g. DMN). Thus, during time windows prior to thought probes, functional connectivity patterns are not only a product of an individual’s level of SITUT but also various other processes such as scene detection. As such, inter-regional interactions involved in SITUT could have been missed due to masking with specific functions associated with ongoing gradCPT performance. Though the SITUT-CPM was sensitive to predictions generated from resting state data, effects were modest, and it remains possible that a different task context for model generation would result in improved out-of-sample prediction performance.

A related concern surrounds our use of 30-second pre-thought probe windows to compute functional connectivity for training and testing of the CPM. This analysis choice was based on our specific task design as well as prior evidence demonstrating that functional connectivity computed from similar durations can reliably distinguish distinct cognitive tasks^{10,11} and mind wandering levels.¹² Though brain activity within the seconds that immediately precede thought probes are most relevant to self-reported experiences,^{3,13} our selection of a ~30-second window was based on the trade-off between the need to capture activity time-locked to experiences and the need to estimate functional connectivity within a window long enough to provide confidence in the sampling of correlations.¹⁴ However, there is no a priori reason to suspect that 30 seconds corresponds to the time course of cognitive processes that unfold during any individual episode of SITUT.¹⁵ In future work, applications of methods sensitive to instantaneous edge co-fluctuations,¹⁶ as well as unsupervised methods sensitive to frame-wise network ‘states’,¹⁷ could be valuable toward better resolving the time course of network dynamics that underlie SITUT.”

2) The authors assess “task-unrelated thought” throughout the gradCPT task, but “external distractions” are assessed only retrospectively at the end of the task. Participants who endorsed external distractions during this retrospective assessment were then excluded from analyses, leaving a sample size of N=17. I think it would be helpful if the authors were to expand on their procedures surrounding this topic, including how many people were excluded from analysis and what rating of external distractions was considered acceptable versus unacceptable enough to warrant inclusion or exclusion. I would also like to see the authors raise the retrospective assessment of external distractions as a study limitation. In particular, participants may not accurately remember what proportion of their stimulus independent thoughts were internal versus external, and other experience sampling studies suggest that stimulus-dependent

distractions are somewhat frequent (e.g. ~25% of probes, Stawarczyk et al., Neural Correlates of Ongoing Conscious Experience: Both Task-Unrelatedness and Stimulus-Independence Are Related to Default Network Activity, PLOS One, 2011).

>>> **RESPONSE:** The reviewer raises important issues regarding our use of retrospective versus online ratings, and we believe that these issues are generally challenging to address during the design of any experience sampling study (especially when combined with neuroimaging). We opted to use retrospective ratings for external distractions, given that extra time would otherwise have been required to conduct such ratings online (along with additional potential items) during scanning. As we aimed to maximize the number of thought probes obtained per participant (given ~1.5 hour scanning sessions), we focused on only two key items (task focus and awareness of task focus) for the online ratings.

Overall, however, we agree that our use of retrospective assessments of external distractions could have been associated with decreased accuracy of ratings, and we certainly recognize that they offer less ‘temporal resolution’ than online ratings would have provided. To partially address this issue, after participants provided retrospective ratings, we asked them to verbally provide specific examples of stimulus-independent thoughts and external distractions that had occurred during task performance. We now noted this in the Methods (Section 2.1) as follows:

“Subsequently, to confirm the numerical ratings, we asked participants to verbally provide specific examples of the stimulus-independent thoughts and external distractions that had occurred during task performance.”

The examples provided by all participants offered us improved confidence in the accuracy of their ratings. However, we appreciate that this does not completely circumvent the problem. Thus, taking the reviewer’s suggestion, we have now also provided further details on the retrospective stimulus-independence (SITUT), external distraction (ED), and task-related interference (TRI) ratings within our selected cohort of 17 participants. Consistent with our prior work,¹⁸ we used the retrospective SITUT, but not ED or TRI, ratings for inclusion criteria. Importantly, in our selected cohort of 17 participants (based on individuals with SITUT ratings ≥ 4), mean SITUT ratings were substantially higher than ED and TRI ratings. Moreover, it was rare for a participant who reported high SITUT to also report high EDs or TRIs (SITUT was negatively correlated with both of these items).

We now note the following in the Results (Section 1.0):

“Specifically, the included participants retrospectively reported that their “off-task” ratings were more strongly due to SITUT ($M \pm SD$ rating on 7-point Likert scale:

5.5±1.1) than to external distractions (EDs: $M \pm SD = 2.1 \pm 1.2$) or to thoughts about task-related interferences (TRIs: $M \pm SD = 2.8 \pm 1.7$) (SITUT vs. EDs: $P = 2.1 \times 10^{-6}$; SITUT vs. TRIs: $P = 1.1 \times 10^{-4}$; Wilcoxon rank sum tests).”

We also note the following in Methods (Section 2.1):

“Across participants, SITUT ratings were negatively correlated with both ED ($\rho = -0.58$, $P = 0.0012$) and TRI (Spearman’s $\rho = -0.47$, $P = 0.011$) ratings, suggesting that it was rare for a participant who reported high SITUT to also report high EDs or TRIs.”

To illustrate the negative relationship between SITUT and ED ratings, we also share here a histogram showing ED ratings in the High SITUT ($n=17$) and Low SITUT ($n=11$) groups:

Finally, we added the following text to the Limitations section of the Discussion:

“In our experience sampling study, online thought probes assessed task focus and awareness of task focus, while retrospective ratings assessed the overall degree to which reports of off-task focus were due to SITUT, external distractions, or task-related interferences. Though online and retrospective ratings typically show good correspondence,¹⁹ retrospective ratings could have been associated with decreased relative accuracy in our study, as participants may not have precisely remembered the proportion of time spent focused off-task for each of the three respective categories. Prior studies, using thought probes to assess these three categories online, suggest that participants may fluctuate from trial to trial in their reasons for reporting off-task focus and that the distinct categories have different neural correlates.²⁰ Moreover, multidimensional experience sampling has revealed that distinct thought categories recur over time and are linked to different functional connectivity patterns across individuals.²¹⁻²³ Future work adopting such a multidimensional approach to online thought probes could better clarify cross-trial heterogeneity within individuals and potentially lead to development of more specific neural models of SITUT components that are either individual-specific or generalize across individuals and populations.”

3) I think it would be informative if the authors were to consider pitting the SITUT-

CPM map and the Sustained Attention-CPM map against each other in the service of examining which map is a better predictor of ADHD presence and symptomatology. The authors could also consider estimating the combined predictive validity from the two maps.

>>> **RESPONSE:** We agree with the reviewer that it would be informative to further explore the clinical relevance of the SITUT-CPM and SA-CPM in ADHD. We therefore have now performed 3 new analyses that we report in the updated manuscript:

(1) In the MGH cohort, which included both ADHD and HC participants, we compared groups in terms of mean SA-CPM network strength during task performance. In Results (Section 2.0), we report the following:

“...SA-CPM network strength was not significantly different between groups [M±SD: ADHD (-12.5±44.6), HC (-25.3±37.5); P = 0.37].”

This result complements the SITUT-CPM analysis that we originally reported in the MGH cohort, where we showed that SITUT-CPM network strength during task performance was significantly greater in ADHD compared to HC participants. Together, the results suggest that the SITUT-CPM is more sensitive than the SA-CPM to ADHD presence (when CPMs are computed during task performance).

(2) In the MIT cohort, which included ADHD participants classified into low or high SITUT subgroups, we examined whether SA-CPM network strength (at rest) was associated with individual differences in MWQ scores. In Results (Section 3.3, we report the following:

“...SA-CPM network strength was not correlated with MWQ score ($\rho = -0.01$, $P = 0.94$).”

This result complements the SITUT-CPM analysis that we originally reported in the MIT cohort, where we showed that SITUT-CPM network strength at rest was associated with individual differences in MWQ score in ADHD. Along with various other results reported in our paper, the finding provides further evidence for specificity of SITUT-CPM-based, relative to SA-CPM-based, predictions of SITUT-related outcomes (in this case, within ADHD patients).

(3) In the MIT cohort, we examined whether SITUT-CPM and SA-CPM network strengths at rest were correlated with individual differences in ADHD symptom scores. These scores included severity of inattention and hyperactivity symptoms, as assessed with Kiddie Schedule for Affective Disorders and Schizophrenia (K-SADS) semi-structured interview (see also response to Reviewer #1 Point #1). We report the results from these analyses in

Supplementary Table 1 (also shown below), and we now summarize the findings in Results (Section 3.3) as follows:

*“We further examined whether SITUT-CPM and SA-CPM network strengths were related to ADHD symptom severity (rather than MWQ scores). As shown in **Supplementary Table 1**, SITUT-CPM network strength was not significantly predictive of symptoms (though trends were positive), while SA-CPM showed its strongest positive relationship with diagnostic inattention severity. These results suggest that SITUT-CPM network strength was more strongly associated with a SITUT-based outcome (MWQ score) than it was with ADHD symptom severity, whereas SA-CPM network strength was more strongly associated with symptom severity (inattention) than with MWQ score.”*

Supplementary Table 1. Associations between CPM-based network strengths (for SITUT-CPM and SA-CPM) and ADHD symptom severity, as assessed based on the Kiddie Schedule for Affective Disorders and Schizophrenia (K-SADS) with separate scores for current (items rated based on the past 6 months) and diagnostic (items rated based on the past in general) symptoms. Spearman’s rank correlation coefficients are shown uncorrected *P* values.

	Current Hyperactivity	Diagnostic Hyperactivity	Current Inattention	Diagnostic Inattention
SITUT-CPM	$\rho = 0.13$ $P = 0.36$	$\rho = 0.17$ $P = 0.25$	$\rho = 0.11$ $P = 0.44$	$\rho = 0.23$ $P = 0.10$
SA-CPM	$\rho = 0.16$ $P = 0.27$	$\rho = 0.12$ $P = 0.41$	$\rho = 0.17$ $P = 0.23$	$\rho = 0.35$ $P = 0.013$

4) It strikes me that the healthy participants might be more likely to deliberately engage in SITUT during the gradCPT task than participants with ADHD (who may do so more spontaneously). If this is the case, I would expect FPCN-DMN connectivity to be stronger in healthy adults if such individuals are deliberately holding their attention on SITUTs. Perhaps the authors could speculate on such a possible source of observed differences in SITUT-CPM between groups? Relatedly, DMN-FPCN connectivity may be higher in healthy individuals if DMN-FPCN connectivity is related to awareness. The authors indicated that they did not analyze the question pertaining to awareness, but doing so may help shed light on the implications of the authors’ findings. Overall, these two hypotheses seem to fit with the authors’ findings that the DMN-FPCN connectivity patterns only significantly predict SITUT in healthy individuals.

>>> **RESPONSE:** This is an excellent point, and we agree that the question that we administered regarding awareness of SITUT could help to shed light on the group difference in DMN-FPCN-based prediction of SITUT. To address this point, we have now compared groups in terms of awareness ratings and found that, as expected, ADHD patients showed lower awareness of their attentional focus than

HC participants. We updated our Methods section to describe inclusion of this analysis, and we now report the findings in Results (Section 2.0) as follows:

“The ADHD group also showed lower awareness of where their attention was focused ($M \pm SD$ on 100-point scale: ADHD (50.3 ± 15.3), HC (67.6 ± 16.7); $P = 0.0016$).”

Within the ADHD group ($n=20$), we also performed a correlation between the success of SITUT-CPM_{DMN-FPCN} performance and mean awareness rating across participants. This revealed a positive trend ($r = 0.25$, $p = 0.29$), providing preliminary support for the idea that DMN-FPCN-based prediction performs better when participants are aware of (and potentially more deliberately engaged in) SITUT. However, the small sample size for this correlation limits interpretability, and so we choose to omit this analysis from the manuscript.

Additionally, we have expanded our Discussion (*Clinical Implications* section) to further speculate that DMN-FPCN coupling corresponded to SITUT in healthy, but not ADHD participants, due to deliberate as opposed to spontaneous SITUT (potentially reflected in the awareness ratings). The new text reads as follows:

“Importantly, although we demonstrated that DMN-FPCN interaction was, overall, a key feature contributing to SITUT-CPM predictions in healthy participants, predictions in ADHD were largely based on other network pairs. A possible reason for this group difference is that ADHD, relative to healthy, participants reported lower awareness of where their attention was focused (in the MGH sample). This decreased awareness may indicate spontaneous, as opposed to deliberate, SITUT, a feature previously linked to ADHD symptoms.^{24,25} It is possible that increased DMN-FPCN coupling more closely reflects deliberate, compared to spontaneous mind wandering,²⁶ and as such, DMN-FPCN coupling corresponded to SITUT only in healthy participants in our study because these individuals were experiencing SITUTs more deliberately than were ADHD participants. Notably, various rs-fMRI studies have shown a relationship between ADHD and reduced resting state DMN anticorrelation with other networks, including the FPCN,²⁷⁻²⁹ as well as with dampened task-evoked FPCN activation and DMN deactivation.³⁰ Thus, we hypothesize that ADHD patients have an altered baseline level of DMN-FPCN interaction (associated with spontaneous SITUT) and that other, distributed network interactions account for psychological features that are common between spontaneous and deliberate SITUT.”

Minor Comments:

5) It would be helpful if the authors were to provide more clarity on their sources for the terms “DMN-A”, “DMN-B”, “DMN-C”, “FPCN-A” and “FPCN-B”. Citations would be helpful here, particularly because the terms “DMN-A” and “DMN-B” have also been used to describe a 2-solution DMN parcellation (see Buckner & DiNicola, “The brain’s default network: updated anatomy, physiology and evolving insights.” *Nature Reviews Neuroscience*, 2019). Do the terms FPCN-A

and FPCN-B come from Dixon et al., “Heterogeneity within the frontoparietal control network and its relationship to the default and dorsal attention networks” PNAS, 2018?

>>> **RESPONSE:** We had accidentally omitted this important information from our manuscript, and we thank the reviewer for pointing out this issue. We have now clarified how we defined these subnetwork terms, as now stated in the Methods as follows:

“To describe our results, we use the labels provided with the publicly available Schaefer atlas (https://github.com/ThomasYeoLab/CBIG/tree/master/stable_projects/brain_parcellation/Schaefer2018_LocalGlobal) with an exception made for FPCN subnetworks found in the 17-network atlas; we flipped the provided FPCN_A and FPCN_B subnetwork labels so that our descriptions are consistent with prior work³¹ in which the subnetwork including rostralateral prefrontal cortex was termed FPCN_A.”

Additionally, in the Results, we cited the Dixon et al. paper when first referring to the FPCN subnetworks.

6) On page 9, the following sentence is a bit confusing. “In summary, although SITUT-CPM predictions were based on a complex, distributed pattern of interacting networks, key components associated with increased SITUT were a) decreased DMNA-FPCNA anticorrelation; and b) increased DMNA-SMNA correlation (Fig. 2g).” Unless I’m misunderstanding, did the authors mean to write “decreased” DMNA-SMNA correlation, rather than “increased” correlation?

>>> **RESPONSE:** We thank the reviewer for noticing this error. Indeed, we meant to write “decreased” here and have now corrected the error.

7) In figure 1d, the transparent lateral brain makes it difficult to visually distinguish between medial versus lateral views, which makes interpretation of the individual regional connectivity patterns a bit challenging. Do the authors have a clearer way to plot lateral and medial views separately?

>>> **RESPONSE:** We appreciate this suggestion and have now updated Figure 1d to include lateral (left and right) as well as top and bottom views that illustrate both lateral and medial nodes and edges (showing high degree nodes only):

Predictive Network Features

We note that visualization remains a general challenge for all studies using connectome-based predictive modeling (discussed in³²). We think that inclusion of these new images offers increased clarity. Additionally, our analyses centered on breaking down the contributions of distinct network pairs (Figure 2) aimed to offer some additional clarity. We have also shared the connectivity matrices with the community (available on Github), and the relevant atlases are also publicly available; thus, researchers may openly explore the contributions of specific nodes and edges to the patterns that we identified.

8) Figure 2g was initially somewhat confusing to me, particularly because I found myself trying to link the background colors “red/pink” to positive predictive links, and “blue” to negative predictive links. Perhaps the authors could choose a different background color scheme.

>>> **RESPONSE:** We thank the reviewer for noticing this inconsistency between the color schemes used for positive versus negative links across Figures 1 and Figure 2 and the various panels. This was an oversight. We have now updated Figure 1d such that positive (high SITUT) links are blue, and the negative (low SITUT) links are red. We have also updated all relevant panels in Figure 2 such that blue and red colors, respectively, are consistently used for SITUT positive and negative directions. As such, the original colors used in Figure 2g are now aligned with all other figures and should not longer lead to confusion.

References

- 1 Biederman, J. *et al.* Clinical correlates of mind wandering in adults with ADHD. *J Psychiatr Res* **117**, 15-23, doi:10.1016/j.jpsychires.2019.06.012 (2019).
- 2 Esterman, M., Noonan, S. K., Rosenberg, M. & Degutis, J. In the zone or zoning out? Tracking behavioral and neural fluctuations during sustained attention. *Cereb Cortex* **23**, 2712-2723, doi:10.1093/cercor/bhs261 (2013).
- 3 Kucyi, A., Esterman, M., Riley, C. S. & Valera, E. M. Spontaneous default network activity reflects behavioral variability independent of mind-wandering. *Proc Natl Acad Sci U S A* **113**, 13899-13904, doi:10.1073/pnas.1611743113 (2016).
- 4 Fortenbaugh, F. C., DeGutis, J. & Esterman, M. Recent theoretical, neural, and clinical advances in sustained attention research. *Ann N Y Acad Sci* **1396**, 70-91, doi:10.1111/nyas.13318 (2017).
- 5 Stawarczyk, D., Majerus, S., Maj, M., Van der Linden, M. & D'Argembeau, A. Mind-wandering: phenomenology and function as assessed with a novel experience sampling method. *Acta Psychol (Amst)* **136**, 370-381 (2011).
- 6 McVay, J. C. & Kane, M. J. Conducting the train of thought: working memory capacity, goal neglect, and mind wandering in an executive-control task. *J Exp Psychol Learn Mem Cogn* **35**, 196-204, doi:10.1037/a0014104 (2009).
- 7 Zanesco, A. P., Denkova, E. & Jha, A. P. Self-reported Mind Wandering and Response Time Variability Differentiate Prestimulus Electroencephalogram Microstate Dynamics during a Sustained Attention Task. *J Cogn Neurosci*, 1-18, doi:10.1162/jocn_a_01636 (2020).
- 8 Maillet, D., Yu, L., Hasher, L. & Grady, C. L. Age-related differences in the impact of mind-wandering and visual distraction on performance in a go/no-go task. *Psychol Aging* **35**, 627-638, doi:10.1037/pag0000409 (2020).
- 9 Esterman, M., Rosenberg, M. D. & Noonan, S. K. Intrinsic fluctuations in sustained attention and distractor processing. *J Neurosci* **34**, 1724-1730, doi:10.1523/JNEUROSCI.2658-13.2014 34/5/1724 [pii] (2014).
- 10 Shirer, W. R., Ryali, S., Rykhlevskaia, E., Menon, V. & Greicius, M. D. Decoding subject-driven cognitive states with whole-brain connectivity patterns. *Cereb Cortex* **22**, 158-165, doi:10.1093/cercor/bhr099 (2012).
- 11 Gonzalez-Castillo, J. *et al.* Tracking ongoing cognition in individuals using brief, whole-brain functional connectivity patterns. *Proc Natl Acad Sci U S A* **112**, 8762-8767, doi:10.1073/pnas.1501242112 (2015).
- 12 Mittner, M. *et al.* When the brain takes a break: a model-based analysis of mind wandering. *J Neurosci* **34**, 16286-16295, doi:10.1523/JNEUROSCI.2062-14.2014 (2014).

- 13 Turnbull, A. *et al.* Reductions in task positive neural systems occur with the passage of time and are associated with changes in ongoing thought. *Scientific reports* **10**, 9912, doi:10.1038/s41598-020-66698-z (2020).
- 14 Leonardi, N. & Van De Ville, D. On spurious and real fluctuations of dynamic functional connectivity during rest. *Neuroimage* **104**, 430-436, doi:10.1016/j.neuroimage.2014.09.007 (2015).
- 15 Kucyi, A. Just a thought: How mind-wandering is represented in dynamic brain connectivity. *Neuroimage* **180**, 505-514, doi:10.1016/j.neuroimage.2017.07.001 (2018).
- 16 Zamani Esfahlani, F. *et al.* High-amplitude cofluctuations in cortical activity drive functional connectivity. *Proc Natl Acad Sci U S A* **117**, 28393-28401, doi:10.1073/pnas.2005531117 (2020).
- 17 Vidaurre, D., Smith, S. M. & Woolrich, M. W. Brain network dynamics are hierarchically organized in time. *Proc Natl Acad Sci U S A* **114**, 12827-12832, doi:10.1073/pnas.1705120114 (2017).
- 18 Kucyi, A. & Davis, K. D. Dynamic functional connectivity of the default mode network tracks daydreaming. *NeuroImage* **100C**, 471-480, doi:S1053-8119(14)00521-7 [pii] 10.1016/j.neuroimage.2014.06.044 (2014).
- 19 Smallwood, J. & Schooler, J. W. The restless mind. *Psychol Bull* **132**, 946-958, doi:2006-20202-006 [pii] 10.1037/0033-2909.132.6.946 (2006).
- 20 Stawarczyk, D., Majerus, S., Maquet, P. & D'Argembeau, A. Neural correlates of ongoing conscious experience: both task-unrelatedness and stimulus-independence are related to default network activity. *PLoS One* **6**, e16997 (2011).
- 21 McKeown, B. *et al.* The relationship between individual variation in macroscale functional gradients and distinct aspects of ongoing thought. *Neuroimage* **220**, 117072, doi:10.1016/j.neuroimage.2020.117072 (2020).
- 22 Turnbull, A. *et al.* Left dorsolateral prefrontal cortex supports context-dependent prioritisation of off-task thought. *Nat Commun* **10**, 3816, doi:10.1038/s41467-019-11764-y (2019).
- 23 Sormaz, M. *et al.* Default mode network can support the level of detail in experience during active task states. *Proc Natl Acad Sci U S A* **115**, 9318-9323, doi:10.1073/pnas.1721259115 (2018).
- 24 Seli, P., Smallwood, J., Cheyne, J. A. & Smilek, D. On the relation of mind wandering and ADHD symptomatology. *Psychonomic bulletin & review*, doi:10.3758/s13423-014-0793-0 (2015).
- 25 Arabaci, G. & Parris, B. A. Probe-caught spontaneous and deliberate mind wandering in relation to self-reported inattentive, hyperactive and impulsive traits in adults. *Scientific reports* **8**, 4113, doi:10.1038/s41598-018-22390-x (2018).
- 26 Christoff, K., Irving, Z. C., Fox, K. C., Spreng, R. N. & Andrews-Hanna, J. R. Mind-wandering as spontaneous thought: a dynamic framework. *Nat Rev Neurosci* **17**, 718-731, doi:10.1038/nrn.2016.113 (2016).

- 27 Sripada, C. S., Kessler, D. & Angstadt, M. Lag in maturation of the brain's intrinsic functional architecture in attention-deficit/hyperactivity disorder. *Proc Natl Acad Sci U S A* **111**, 14259-14264, doi:10.1073/pnas.1407787111 (2014).
- 28 Mattfeld, A. T. *et al.* Brain differences between persistent and remitted attention deficit hyperactivity disorder. *Brain* **137**, 2423-2428, doi:10.1093/brain/awu137 (2014).
- 29 Castellanos, F. X. *et al.* Cingulate-precuneus interactions: a new locus of dysfunction in adult attention-deficit/hyperactivity disorder. *Biol Psychiatry* **63**, 332-337, doi:10.1016/j.biopsych.2007.06.025 (2008).
- 30 Hart, H., Radua, J., Nakao, T., Mataix-Cols, D. & Rubia, K. Meta-analysis of functional magnetic resonance imaging studies of inhibition and attention in attention-deficit/hyperactivity disorder: exploring task-specific, stimulant medication, and age effects. *JAMA psychiatry* **70**, 185-198, doi:10.1001/jamapsychiatry.2013.277 (2013).
- 31 Dixon, M. L. *et al.* Heterogeneity within the frontoparietal control network and its relationship to the default and dorsal attention networks. *Proc Natl Acad Sci U S A* **115**, E1598-E1607, doi:10.1073/pnas.1715766115 (2018).
- 32 Shen, X. *et al.* Using connectome-based predictive modeling to predict individual behavior from brain connectivity. *Nat Protoc* **12**, 506-518, doi:10.1038/nprot.2016.178 (2017).

Reviewers' Comments:

Reviewer #1:

Remarks to the Author:

The authors have addressed all of my points well -- thank you.

Reviewer #2:

Remarks to the Author:

The authors have done an excellent job at addressing my concerns, and I have no further comments. I think this manuscript will make a fine contribution to the literature.